# Dependency of Particle Size Distribution at Dust Emission on Friction Velocity and Atmospheric Boundary-Layer Stability

Yaping Shao[1], Jie Zhang[2], Masahide Ishizuka[3], Masao Mikami[4], John Leys[5, 6], Ning Huang[2]

[1] Institute for Geophysics and Meteorology, University of Cologne, Germany
[2] Key Laboratory of Mechanics on Disaster and Environment in Western China, Lanzhou University, China
[3] Faculty of Engineering and Design, Kagawa University, Japan
[4] Office of Climate and Environmental Research Promotion, Japan Meteorological Business Support Center, Japan
[5] Department of Planning, Industry and Environment, New South Wales, Australia
[6] The Fenner School of Environment & Society, The Australian National University, Australia

*Correspondence to*: Jie Zhang (zhang-j@lzu.edu.cn) and Ning Huang (huangn@lzu.edu.cn)

**Abstract.** Particle size distribution of dust at emission (dust PSD) is an essential quantity to estimate in dust studies. It has been recognized in earlier research that dust PSD is dependent on soil properties (e.g. whether soil is sand or clay) and friction velocity, $u_*$, a surrogate for surface shear stress and descriptor for saltation bombardment intensity. This recognition has been challenged in some recent papers, causing a debate on whether dust PSD is "invariant" and the search for its justification. In this paper, we analyse the dust PSD measured in the Japan-Australian Dust Experiment and show that dust PSD is dependent on $u_*$ and on atmospheric boundary-layer (ABL) stability. By simple theoretical and numerical analysis, we explain the two reasons for the latter dependency, both related to enhanced saltation bombardment in convective turbulent flows. First, $u_*$ is stochastic and its probability distribution profoundly influences the magnitude of the mean saltation flux due to the non-linear relationship between saltation flux and $u_*$. Second, in unstable conditions, turbulence is usually stronger, which leads to higher saltation-bombardment intensity. This study confirms that dust PSD depends on $u_*$, and more precisely, on the probability distribution of $u_*$, which in turn is dependent on ABL stability, and consequently dust PSD is also dependent on ABL. We also show that the dependency of dust PSD on $u_*$ and ABL stability is made complicated by soil surface conditions. In general, our analysis reinforces the basic conceptual understanding that dust PSD depends on saltation bombardment and inter-particle cohesion.

## 1 Introduction

Gillette (1981) explained that dust emission can be produced by aerodynamic lift and saltation bombardment, but under realistic wind, aerodynamic-lift emission is much weaker than saltation-bombardment emission. This hypothesis was confirmed by Shao et al. (1993). It is recognized that saltation bombardment is the most important mechanism for dust emission, and dust emission rate, $F$, is proportional to streamwise saltation flux, $Q$[1].

Rice et al. (1995, 1996) visualized the process of saltation bombardment using wind-tunnel photos: a saltation particle at impact on surface ejects a tiny amount of soil into air, leaving behind a crater. Models for estimating crater size have been developed by, e.g., Lu and Shao (1999). The fraction of dust that gets emitted from the ejection is difficult to estimate, because it depends both on inter-particle cohesion and bombardment intensity. Since inter-particle cohesion depends on particle size, $d$, the fraction of dust emitted must also depend on $d$. Thus, for a given soil, the particle size distribution of dust at emission (emission-dust PSD), $p_s(d)$, must depend on saltation bombardment or on friction velocity, $u_*$ ($\sqrt{\tau}/\rho$ with $\tau$ being surface shear stress and $\rho$ air density; see Section 4.2 for discussion). Alfaro et al. (1997) confirmed that $p_s(d)$ depends on $u_*$: as $u_*$ increases, $p_s(d)$ shows a higher fraction of dust of smaller $d$. Based on this result and the observation that different laboratory techniques for PSD analysis yield profoundly different outcomes, depending on the disturbances applied to the samples (Figure 1), Shao (2001) suggested to use a minimally-disturbed PSD, $p_m(d)$, as the limit of $p_s(d)$ for weak saltation, and a fully-disturbed PSD, $p_f(d)$, as the limit of $p_s(d)$ for strong saltation. In this way, $p_s(d)$ is expressed as a weighted average of $p_m(d)$ and $p_f(d)$

$$p_s(d) = \gamma p_m(d) + (1 - \gamma)p_f(d), \tag{1}$$

where $0 \leq \gamma \leq 1$ is an empirical function of $u_{*t}(d)$, the threshold friction velocity for particles of size $d$.

What is emission-dust PSD? We distinguish three closely related yet subtly different dust PSDs, namely, emission-dust PSD, airborne-dust PSD, and emission-flux PSD. PSD of dust suspended in air (airborne-dust PSD) has been collected from different places under different conditions. Emission-dust PSD and airborne-dust PSD are identical, if the latter is measured at dust source at height zero. Airborne-dust PSD can be used to approximate emission-dust PSD if it is measured close to the source and the dependency of particle motion in air on particle size can be neglected. For modelling size-resolved dust concentration in air (i.e. solving the dust concentration equation for different particle sizes), emission-dust PSD offers the Dirichlet-type boundary condition. If size-resolved dust-emission-fluxes can be calculated, then we can specify the Neumann-type boundary condition for solving the dust concentration equation. From size-resolved dust-emission-fluxes, an emission-flux PSD can be calculated (Section 2; Section 4.2). Emission-flux PSD is neither emission-dust nor airborne-dust PSD, but describes how vertical dust-concentration gradient depends on particle size. In some earlier publications, unfortunately, the differences between the three dust PSDs are not clearly explained.

---

[1]The ratio $\gamma_b = F/Q$ is a main issue in dust emission studies (Zender et al., 2003; Laurent et al., 2006). Marticorena et al. (1997) showed that $\gamma_b$ depends on soil clay content. Shao (2004) suggested that $\gamma_b$ depends on friction velocity, soil type and soil particle size distribution.

To our knowledge, emission-dust PSD has never been directly measured, but approximated using airborne-dust PSD
measured at some, often different, heights (e.g. Kok, 2011b, Table S1). Available data of airborne-dust PSDs give the
impression that they do not differ much. It has thus been suggested that airborne-dust PSDs may be "not-so-different" and
hence emission-dust PSDs may also be "not-so-different". Reid et al. (2008) stated that "on regional scales, common mode
dust is not functionally impacted by production wind speed, but rather influenced by soil properties such as
geomorphology …". Kok (2011a, 2011b) proposed a dust emission model by treating dust emission as a process of
aggregate fragmentation by saltation bombardment. Since aggregate fragmentation is similar to brittle fragmentation, the size
distribution produced in the process is scale-invariant (Astrom, 2006). Kok (2011a, 2011b) then proposed an emission-dust
PSD and estimated its parameters from the data listed in Table S1 of Kok (2011b). The proposed emission-dust PSD is
frequently used in dust models (Giorgi et al., 2012; Albani et al., 2014; Pisso et al., 2019). However, whether the "not-so-
different" airborne-dust PSDs justify "brittle fragmentation" as the underlying process for dust emission requires scrutiny.
Studies on dust PSD are yet to deliver definitive answers. The airborne-dust PSD measurements of Rosenberg et al. (2014)
pointed to larger fraction of fine particles than in earlier published data. Ishizuka et al. (2008) found that airborne-dust PSD
measured close to surface depends on $u_*$ for a weakly crusted soil. Sow et al. (2009) examined the dependency of emission-
flux PSD on $u_*$ for three dust events and reported that the PSD appeared to be independent on $u_*$, but differed significantly
between weak and strong events. In line with Sow et al. (2009), Khalfallah et al. (2020) reported that emission-flux PSD
depends on atmospheric boundary-layer (ABL) stability, and attributed this to the dependency of particle diffusivity on
particle size. They stated that the dependency of emission-dust PSD on $u_*$, as observed by Alfaro et al. (1997), may be of
secondary importance in natural conditions compared to its dependency on ABL stability.
The argument of Khalfallah et al. (2020) rests on the preferential particle diffusion in turbulent flows. Csanady (1963)
suggested that particle eddy diffusivity, $K_p$, is related to eddy diffusivity, $K$, by

$$K_p = K(1 + \beta^2 w_t^2/\sigma^2)^{-1/2}, \tag{2}$$

where $\beta$ is a coefficient, $w_t$ particle terminal velocity and $\sigma$ the standard deviation of (vertical) turbulent velocity. The
analyses of Walklate (1987) and Wang and Stock (1993), among many others, reached similar conclusions. For dust particles
smaller than 10μm, $K_p/K$ is close to one for $\sigma = 0.5$ ms$^{-1}$, and still larger than 0.95 for $\sigma = 0.1$ ms$^{-1}$ (Shao, 2008; Fig. 8.12).
Thus, preferential particle diffusion does not seem to fully explain the dependency of dust PSD on ABL stability.
The confusion ground emission-dust PSD prompted us to re-examine the data of Ishizuka et al. (2008) from the Japan-
Australian Dust Experiment (JADE). In JADE, airborne-dust PSD were measured at small height directly above the dust
source and can be assumed to well approximate the emission-dust PSD. By composite analysis for different $u_*$ and ABL
stabilities, we show that dust PSD depends on $u_*$, supporting the findings of Alfaro et al. (1997), and depends on ABL
stability, consistent with the findings of Khalfallah et al. (2020). But in contrast to Khalfallah et al. (2020), we argue that
these dependencies are not mutually exclusive, but collectively point to the simple physics that emission-dust PSD is
dependent on saltation-bombardment intensity and efficiency.

## 2 JADE Data

JADE was carried out during 23 Feb ~ 14 Mar 2006 on an Australian farm at (33°50'42.4"S, 142°44'9.0"E) (Ishizuka et al., 2008, 2014). The 4 km² farmland was flat and homogeneous such that the JADE data are not affected by fetch. In JADE, atmospheric variables, land surface properties, soil PSD, size-resolved sand fluxes and dust concentrations were measured. Size-resolved dust-emission fluxes were estimated from the dust concentration measurements. Three Sand Particle Counters (SPCs) (Mikami et al., 2005) were used to measure the sand fluxes in the size range of 39 - 654 µm in 32 bins at 0.05, 0.1 and 0.3 m above ground at a sampling rate of 1 Hz. Using the sand fluxes, $q_j$ ($j$ = 1, 32), the PSD of saltation particles (saltation-flux PSD) is estimated for a particle size bin at $d_j$ with bin size $\Delta d_j$ as

$$p(d_j)\Delta d_j = q_j / \sum_{j=1}^{j=32} q_j \qquad (2)$$

Dust concentration was measured using Optical Particle Counters (OPC) for 8 size groups: 0.3 – 0.6, 0.6 – 0.9, 0.9 – 1.4, 1.4 – 2.0, 2.0 – 3.5, 3.5 – 5.9, 5.9 – 8.4 and 8.4 – 12.0 µm at 1, 2 and 3.5 m above ground. The upper size limit for the last bin is not well defined, but set empirically to 12.0µm such that this bin can still be included in the analysis. Airborne-dust PSD is estimated as

$$p(d_j)\Delta d_j = c_j / \sum c_j, \qquad (3)$$

where $c_j$ denotes the dust concentration for size bin $j$. Similarly, the emission-flux PSD can be defined as

$$p(d_j)\Delta d_j = F_j / \sum F_j, \qquad (3a)$$

where $F_j$ denotes the dust flux for size bin $j$. It should be noted that the emission-flux PSD describes how the covariance of particle-velocity and particle-concentration depends on particle size, not the concentration itself. In this study, we use the airborne-dust PSD observed at 1 m to approximate emission-dust PSD, and use the airborne-dust PSD observed at 3.5 m and the emission-flux PSD derived from the 3.5 m- and 1 m-OPC measurements for additional discussions (Section 4.2). Hereafter, emission-dust PSD approximated using the 1m-OPC airborne-dust PSD is simply referred to as dust PSD, unless otherwise stated.

Atmospheric variables, including wind speed, air temperature and humidity at various levels, radiation and precipitation were measured using an automatic weather station. These quantities were sampled at 5-second intervals and their 1-minute averages were recorded (see Section 4.2 for discussions). Two anemometers mounted at 0.53 and 2.16 m measured wind speed. From the atmospheric data, the Obukhov length, $L$, sensible heat flux, $H$, and friction velocity, $u_*$, were derived.[2] Also measured were soil temperature and moisture.

---

[2] Drag-partition theory (Raupach, 1992; Webb et al., 2019) tells that shear stress, $\tau = \rho u_*^2$, is not the same as the shear stress, $\tau_s$, experienced by soil particles, due to roughness sheltering. For JADE, the surface is bare and thus the effect of roughness sheltering is neglected. The saltation fluxes used in this study are measured and do not involve the assumption $\tau = \tau_s$ or otherwise.

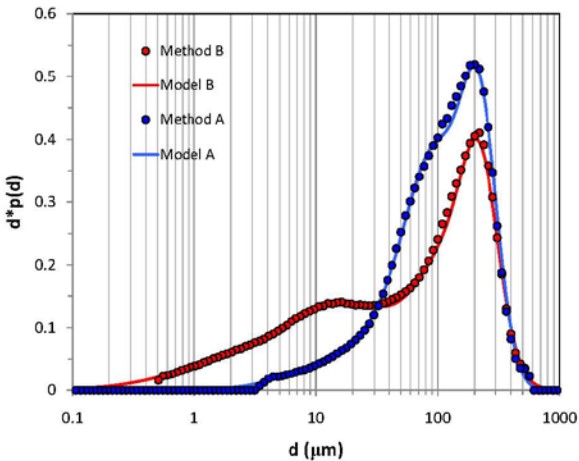


Figure 1. Soil particle-size distribution obtained using Method A and Method B, together with the respective approximations (Model A
and Model B).
Surface soil samples were taken and soil PSD was analysed in laboratory using Method A and B with a particle size
analyser (Microtrac MT3300EX, Nikkiso). In Method A, water was used for sample dispersion with no ultrasonic action. In
Method B, sodium hexametaphosphate (HMP) 0.2% solution was used for sample dispersion and 1-minute ultrasonic action
of 40 W was applied. Following the convention of sedimentology, the soil is a sandy loam based on the analysis using
Method B. Figure 1 shows $p_A(d)$ (soil PSD from Method A) and $p_B(d)$ (soil PSD from Method B) and the corresponding
approximations: $p_A$ shows a larger fraction of particles in the range of 30~300 µm, while $p_B$ a larger fraction of particles in
the range of 0.1~30 µm.
An overview of the JADE data is shown in Figure 2. During the experiment, 12 significant aeolian events were recorded,
as marked in the figure. Most of the events occurred under unstable ABL conditions. Several quantities can be used as a
measure of ABL stability, but the one used here is the convective scaling velocity, $w_*$, defined as
$$w_* = \left(\frac{g}{\bar{\theta}} H_0 z_l\right)^{\frac{1}{3}},$$
(4)

where $g/\bar{\theta}$ is the buoyancy parameter with $g$ being the acceleration due to gravity and $\bar{\theta}$ the mean potential temperature; $H_0$ is
surface kinematic heat flux (K m s$^{-1}$) and $z_l$ a scaling length (set to the capping inversion height for convective ABL and 100
m for stable ABL). For unstable conditions, $w_*$ is positive while for stable conditions $w_*$ is negative. The reason for choosing
$w_*$ is that it is a scaling parameter for the strength of turbulence. Usually, $w_*$ is not used for stable ABLs, but used here as an
indicator for the suppression of turbulence by negative buoyancy.
In addition to the 12 events, a number of weak and intermittent events occurred. In this study, we first use the whole
dataset for the dust PSD analysis, and then use the data for Event-10, 11 and 12 for case studies. These three events are
chosen for that Event-10 is the strongest event during JADE, Event-11 is one that occurred at night under stable conditions,
while Event-12 occurred with a weakly crusted soil surface (Ishizuka et al., 2008).

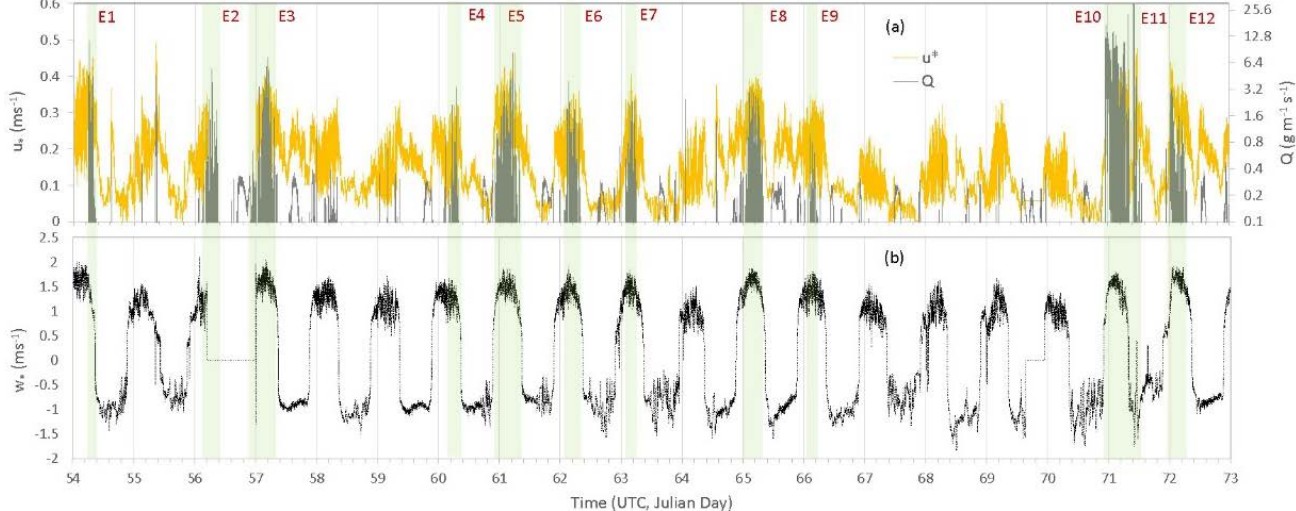


Figure 2. (a) One-minute averaged friction velocity, $u_*$, and streamwise saltation flux, $Q$, for the JADE observation time period; (b) One-
minute averaged convective scaling velocity, $w_*$. In addition to the 12 aeolian events marked, a number of weaker and intermittent aeolian
events occurred.
**3 Results**
**3.1 Overall Results**

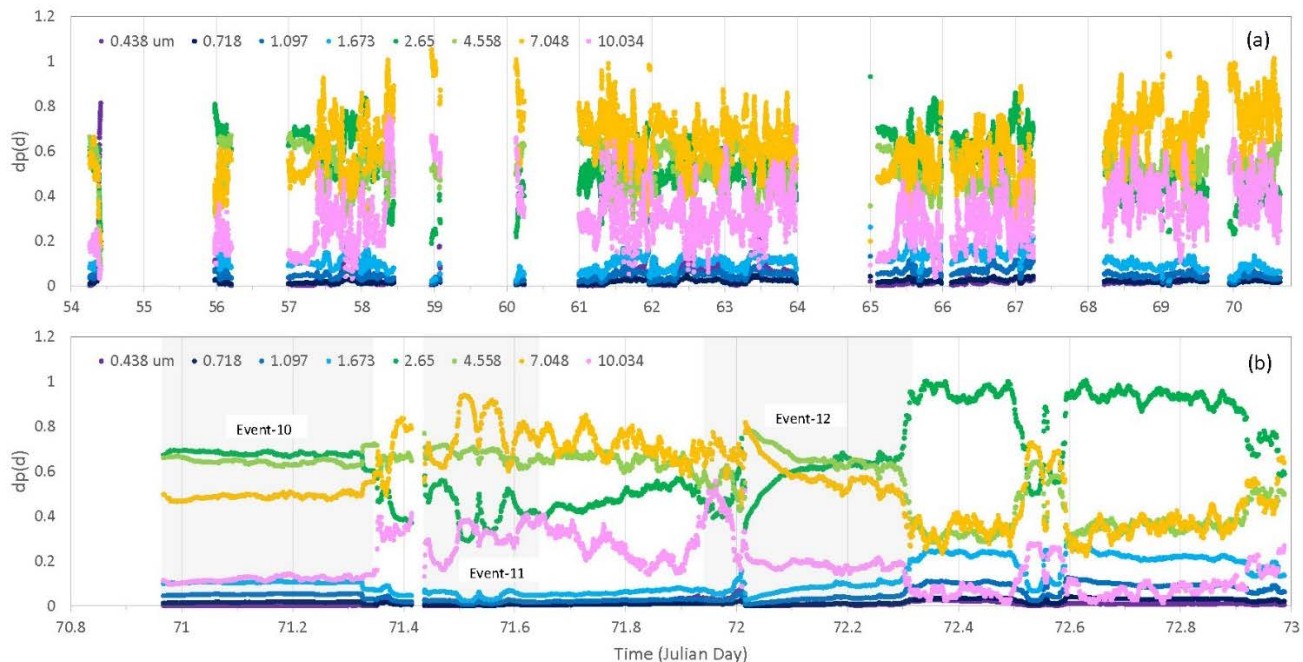

Figure 3. Dust PSD measured at 1m using OPC for the entire JADE observation period plotted in two sections, (a) for section Julian day 54 ~ 70.8 and (b) for section Julian day 70.8 ~ 73.0.

Plotted in Figure 3 are the time series of dust PSD for the entire JADE period, which show rich temporal variations, probably apart from Event-10. To examine dust-PSD dependency on friction velocity, we use $u_*$ to denote the one-minute values of friction velocity, $p(u_*)$ its probability density function (PDF), $\bar{u}_*$ its mean and $\sigma_{u_*}$ its standard deviation. The $u_*$ values are divided into the categories of 0~0.25, 0.25~0.35, 0.35~0.45 and 0.45~0.55 ms$^{-1}$, and the corresponding dust PSDs and saltation PSDs are sorted accordingly. These $u_*$ categories correspond roughly to intermittent, weak, moderate and strong saltation, respectively. The threshold friction velocity, $u_{*t}$, for the JADE site is around 0.2 ms$^{-1}$, but intermittent saltation has been observed oft at $u_*$ below this $u_{*t}$. The dust PSDs are then composite averaged for the $u_*$ categories. Figure 4a shows the dust PSDs for the different $u_*$ categories and the mean dust PSD averaged over all $u_*$ categories (including a total of 15634 one-minute points). We have repeated the same averaging procedure using a subset of the JADE data, conditioned with $Q > 0.1$ gm$^{-1}$s$^{-1}$ and found that the results are very similar to those presented in Figure 4. The mean dust PSD shows an interesting local minimal at ~ 4 μm. This is attributed to the lack of particles of this size in the $u_* < 0.25$ ms$^{-1}$ category. Figure 4a shows that dust PSD clearly depends on $u_*$, particularly in the size range 2 ~ 10 μm. In general, as $u_*$ increases, the fraction of fine dust particles increases. For the submicron size range, the dependency of dust PSD on $u_*$ is less definitive. The dust PSD for the $u_* < 0.25$ ms$^{-1}$ category shows a higher fraction of submicron dust particles, especially in stable conditions (Figure 4b). Apart from this, the results shown in Figure 4a are consistent with the findings of Alfaro et al. (1997) that dust PSD is $u_*$ dependent.

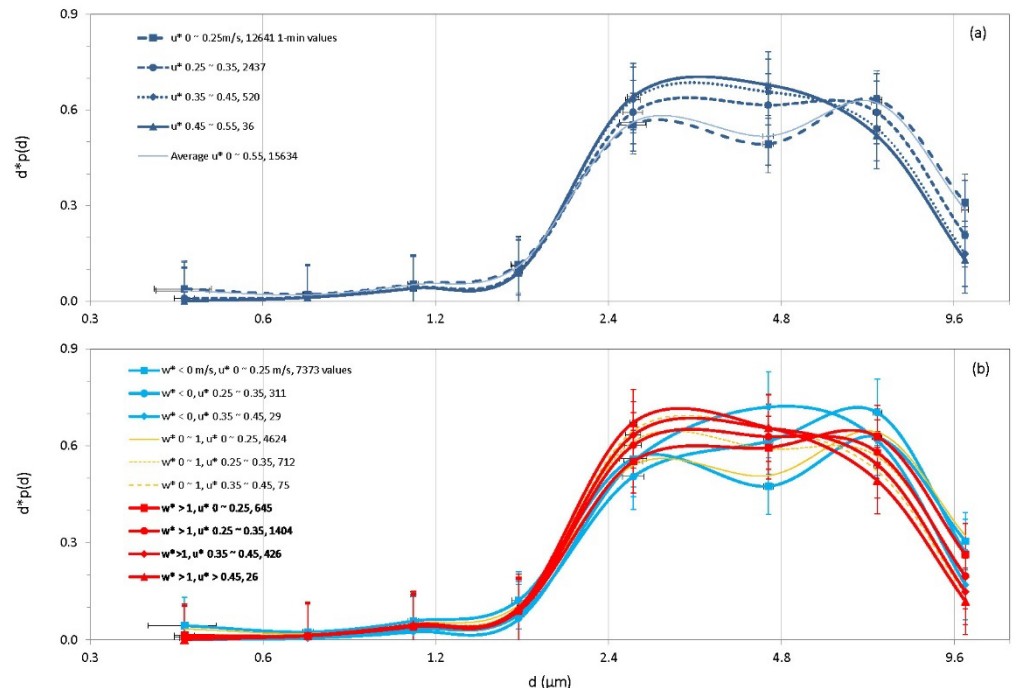


Figure 4. (a) Dust PSD for different $u_*$ categories derived from the whole JADE dataset; (b) as (a), but for the different $u_*$
categories under stable ($w_* < 0$), moderately unstable ($0 \leq w_* < 1$ ms$^{-1}$) and unstable ($w_* \geq 1$ ms$^{-1}$) conditions.

170        To examine the dust PSD dependency on ABL stability, we divide the dataset into stable ($w_* < 0$), moderately unstable ($0$

$\leq w_* < 1$ ms$^{-1}$) and unstable ($w_* \geq 1$ ms$^{-1}$) stability classes. For each stability class, the dust PSD data are regrouped according
to the $u_*$ categories. Figure 4b shows the dust PSDs averaged for different $u_*$ categories and stability classes. For given
stability class, dust PSD shows dependency on $u_*$, and for a given $u_*$ category, dust PSD shows dependency on $w_*$. For given
$u_*$, the mode of dust PSD shifts systematically to finer particles as the ABL becomes more unstable.

**3.2 Case Study Results**

177        We now study the cases of Event-10 (09:49~19:13 12 Mar 2006; Julian Day 70.9506940~71.3423611), Event-11 (21:12

12 Mar ~ 02:08 13 Mar 2006, Julian Day 71.42500~71.63056) and Event-12 (09:54~18:58 13 Mar 2006, Julian Day
71.95417~72.33194). Figure 5 shows the one-minute averages of wind speed at 0.53 m, $U$, air temperature at 0.66 m, $T$,
saltation flux at 0.05 m, $q_{5cm}$, and dust concentration (summed over all particle size bins) at 1 m, $C_{1m}$. Event-10 occurred
under daytime unstable conditions. It was a very hot day prior to a cool change (cold front causing temperature drop but no
rainfall), with near surface air temperature reaching 52$^{\text{o}}$C and wind speed ~8 ms$^{-1}$. The event lasted ~10 hours. The cool
change occurred at ~19:00-21:00 13 Mar 2006 local time. While precipitation was not recorded by the rain gauge (with
resolution of 0.2 mm), the rain sensor [PPS-01(C-PD1), PREDE Co. Ltd.], as marked in Figure 5b, sensed an event of
raindrops shortly before the cool change, lasting about two minutes, and shortly after, lasting about one minute (Ishizuka et
al., 2008). The strong winds (probably also strong sand drift and dust emission) accompanying the cool change caused the
shutdown of the instruments and thus, unfortunately, this period was not fully recorded. Event-11 occurred under stable
conditions after the cool change in the night time of 12/13 Mar 2006, during which $T$ was dropping from ~40$^{\circ}$C to ~33$^{\circ}$C and
$U$ from ~8 ms$^{-1}$ to ~5 ms$^{-1}$. Event-11, which can also arguably be considered to be part of Event-10, was much weaker than
Event-10.
As the OPC measurements were taken close to the surface and directly above the dust source, the dust-concentration
values were generally high. For Event-10, the mean, standard deviation, maximum and minimum of $C_{1m}$ are respectively
7.56, 8.56, 65.96 and 0.02 mg m$^{-3}$, and for Event-11 3.05, 10.57, 100.17 and 0.04 mg m$^{-3}$. The extremely high dust
concentrations measured shortly before and after the cool change could be affected by dust advection and are excluded from
the analysis (although their inclusion made no difference to the event averages of the dust PSDs). For other times, it can be
safely assumed that the dust observed was locally emitted.

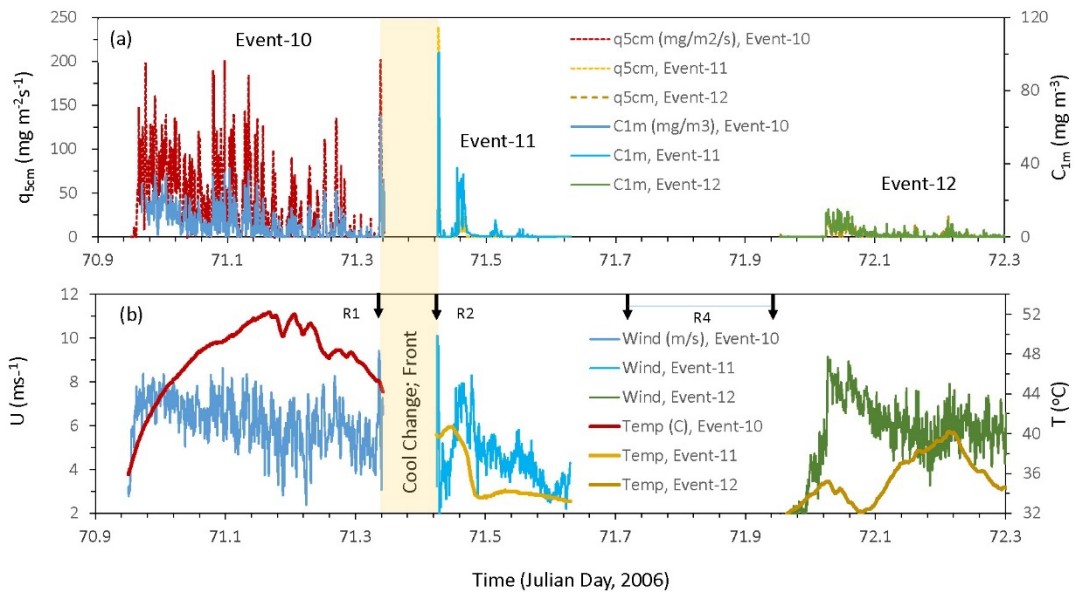

Figure 5. (a) one-minute averaged saltation flux at 0.05 m, $q_{5cm}$, and dust concentration at 1 m, $C_{1m}$, for Event-10, -11 and 12;
(b) as (a) but for wind speed at 0.53 m above ground, $U$, and air temperature at 0.66 m, $T$. The cool change is marked and the
three rain events sensed by the rain sensor are marked as R1, R2 and R4 using the black arrows.

Event-12 is developed shortly after the weak rainfall event (R4). Again, while precipitation was not recorded by the rain
gauge (i.e. the total rainfall was less than 0.2 mm), the rain sensor reported rain drops during 71.70625~71.95278. Ishizuka
et al. (2008) reported that Event-12 is unique for JADE, because it is the only case when the soil surface was weakly crusted.
We will show later how dust PSD can substantially evolve even within one dust event, as soil surface conditions change
(Figure 10).

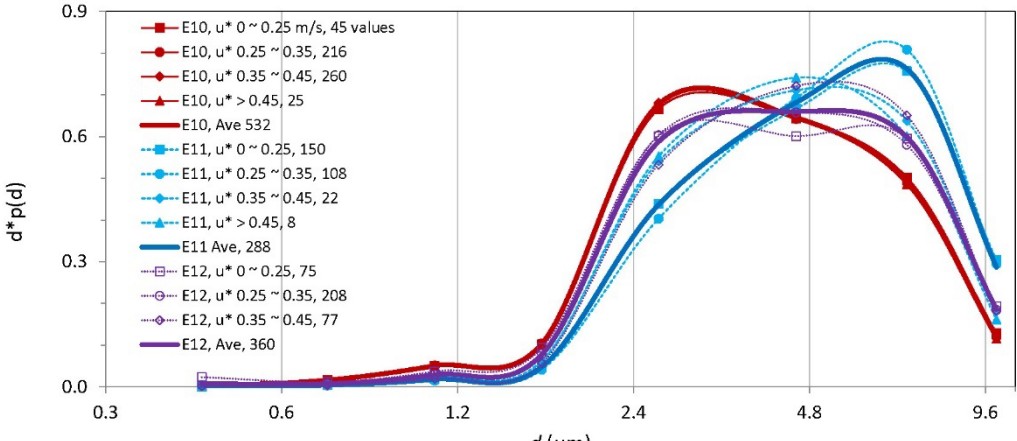


Figure 6. Dust PSD for different $u_*$ categories for Event-10, 11 and 12. Also shown are the PSDs averaged over all $u_*$ categories for the
individual events.
Figure 6 shows the dust PSDs for the different $u_*$ categories for Event-10, 11 and 12. For Event-11 and 12, the
dependency of dust PSD on $u_*$ is obvious, in agreement with the overall results shown in Figure 4a. The dust PSD for Event-
10 shows no clear dependency on $u_*$, which was reported in Shao et al. (2011). Our basic argument for dust PSD dependency
on $u_*$ rests upon the assumption that saltation-impact speed is $u_*$ dependent. It has been suggested that impact-particle speed
may not strongly depend on $u_*$ for transport-limited saltation (Ungar and Haff, 1987), because particle-flow feedbacks force
an approximately constant saltation-impact speed. While this argument is supported by some experimental evidence (Martin
and Kok, 2017) and numerical simulations (Duran et al., 2012; Kok et al., 2012), its general validity and the conditions for
its validity need further examination. JADE Event-10 is probably a case which comes closest to meet the requirements of
strong particle-flow feedback and sustained equilibrium of saltation for the Ungar and Haff (1987) hypothesis to apply. In
addition, Event-10 occurred on an extremely hot and dry day, with the 0.66 m air temperature reaching ~52$^{\circ}$C and relative
humidity below 3%. It is likely that under such extreme conditions, inter-particle cohesion is destroyed. These factors
combined may be responsible for the lack of dust PSD dependency on $u_*$ for Event-10 (Figure 6). But for all other JADE
events, the dependency of dust PSD on $u_*$ is significant.
The event-averaged dust PSDs for Event-10, -11 and -12 clearly differ. For Event-10, the mean and standard deviation of
$u_*$ and $w_*$ were respectively (0.36, 0.057) and (1.03, 0.29), all in ms$^{-1}$, and for Event-11 (0.28, 0.077) and (-0.41, 0.159).
From Event-10 to -11, the dust PSD mode shifted from about 3 µm to 6 µm. During Event-10, a substantially higher fraction
of particles in the size range of 0.4 ~ 4 µm existed. To further examine how dust PSD depends on saltation intensity, we have
averaged the dust PSDs for different $Q$ categories (not shown). It is found that weak saltation corresponded to coarser dust
particles and strong saltation to finer dust particles. Figure 6 confirms the dependency of dust PSD on ABL stability,
consistent with the overall results shown in Figure 4.
Figure 5b shows that the wind conditions for Event-10 and Event-12 were not too different, but Event-12 was much
weaker. Figure 6 shows that also the dust PSDs for the two events considerably differ, with Event-10 being the one with
richer finer dust particles. Event-12 will be further discussed in Section 4.2.
We make the following observations based on the JADE data: (1) Dust PSD has rich temporal variations and is not
"universal"; (2) Dust PSD depends on $u_*$ and ABL stability; and (3) Dust PSD is influenced by soil surface conditions. These
observations support the conceptual understanding that dust PSD is determined both by saltation bombardment and by soil
binding strength (Shao, 2001, 2004).
**4 Discussions**
**4.1 Influence of Turbulence on dust PSD**
The reason for the dependency of dust PSD on $u_*$ has been explained in Gillette et al. (1974), Gillette (1981), Shao et al.
(1993), Alfaro et al. (1997) and Shao (2001; 2004), because $u_*$ is a descriptor of saltation bombardment intensity. In the
earlier explanations, only mean friction velocity and mean saltation are considered, while the turbulent nature of saltation
bombardment is implicitly neglected. But how is the dependency of dust PSD on ABL stability, here $w_*$, explained? The
most conspicuous reason is the enhanced saltation bombardment by turbulence in unstable conditions.

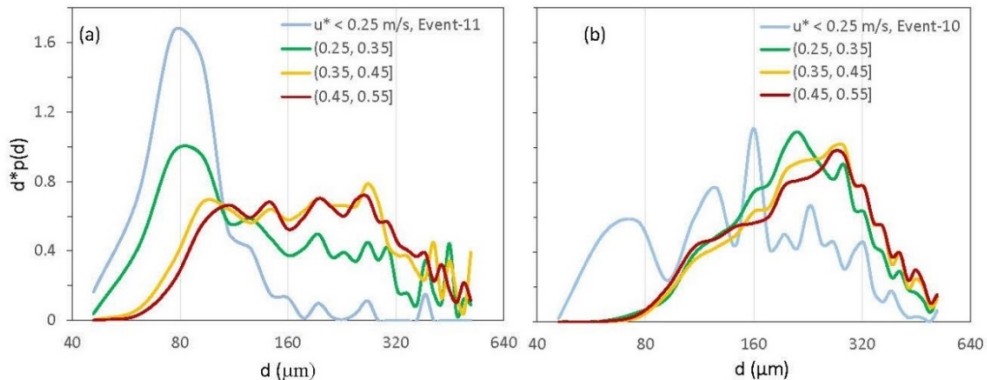


Figure 7. (a) Saltation PSD averaged for four different $u_*$ categories for Event-11; (b) as (a), but for Event-10.
It is interesting to examine how dust PSD is related to saltation PSD. The saltation PSD for Event-10 and -11 are shown in
Figure 7. First, for $u_* \leq 0.25$ ms$^{-1}$ in Event-11, saltation PSD was confined to a narrow size range centred at 70~80 µm where
$u_{*t}$ is minimum. This indicates that saltation splash/bombardment was weak to mobilize particles in other size ranges. In
contrast, for $u_* \leq 0.25$ ms$^{-1}$ in Event-10, saltation PSD covered a broader size range, implying that saltation splash was strong
to entrain particles of other sizes. Second, for both Event-10 and -11, the peak values of saltation PSD were shifted to larger
particles for larger $u_*$: for Event-10 the peak for $0.25 < u_* \leq 0.35$ ms$^{-1}$ was at 203.3 μm, while for $0.45 < u_* \leq 0.55$ ms$^{-1}$ at
257.8 μm. Clearly, since $u_{*t}$ is particle size dependent, saltation PSD is a selective sample of the soil PSD by wind. Third, the
saltation PSDs for given $u_*$ categories (e.g., $0.35 < u_* \leq 0.45$ ms$^{-1}$, Figure 8a and 8b) differed significantly between Event-10
and -11 as a consequence of ABL stability. In Event-11 (Figure 8a), saltation was not fully developed, as the saltation PSD
plateau in the size range 100~300 μm suggests, implying that saltation splash/bombardment was not efficient. In Event-10
(Figure 7b), saltation was more fully developed.
The stronger saltation of Event-10 is partially attributed to the stronger wind and instability, which result in a larger $\bar{u}_*$
than in Event-11. It is known from the ABL similarity theory that,
$$\bar{u}_* = \frac{kz}{\phi_m}\frac{\partial \bar{u}}{\partial z},$$
(5)

where $\kappa$ is the von Karman constant, $z$ height and $\phi_m$ a similarity function (Stull, 1988):
$$\phi_m = \begin{cases} 1 + \beta_m \zeta & \zeta > 0 \text{ stable} \\ (1 - \gamma_m \zeta)^{-1/4} & \zeta < 0 \text{ unstable}, \\ 1 & \zeta = 0 \text{ neutral} \end{cases}$$
(6)

where $\zeta = z/L$ ($L$ is Obukhov length) and $\beta_m = 5$ and $\gamma_m = 16$ are empirical coefficients (Businger et al., 1971). For stable
conditions, $\phi_m > 1$ and for unstable conditions $\phi_m < 1$. Figure 8 shows the PDFs of $u_*$ and $w_*$ for Event-10 and -11,
together with the approximations for the PDFs of $u_*$. For Event-10, $\bar{u}_* = 0.37$ ms$^{-1}$, while for Event-11, $\bar{u}_* = 0.28$ ms$^{-1}$.

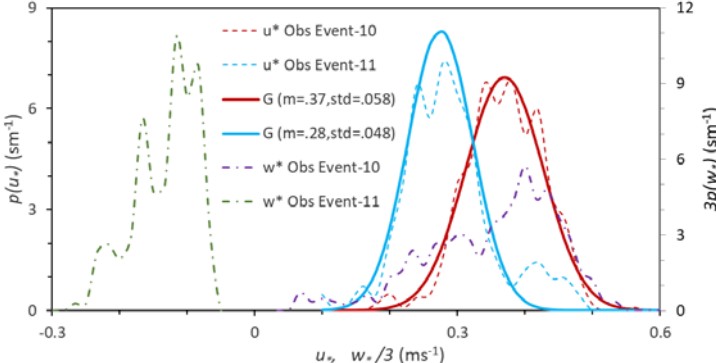


Figure 8. The probability density functions of $u_*$ and $w_*$, $p(u_*)$ and $p(w_*)$, respectively, for Event-10 and -11, together with the Gaussian
approximations for the $p(u_*)$ functions. The mean values (m) and standard deviations (std) for the Gaussian (G) distributions are given.
Note that for $p(w_*)$, $3p(w_*)$ against $w_*/3$ is plotted to conveniently present the information in the same graph.
We suggest that the dependency of dust PSD on $w_*$ for given $u_*$ is attributed to saltation bombardment intensity from two
perspectives. First, as Figure 8 shows, $u_*$ is a stochastic variable. Li et al. (2020) suggested that $\tau = \rho u_*^2$ in neutral conditions
is Gauss distributed. Klose et al. (2014) reported that $\tau$ in unstable conditions is Weibull distributed. The exact form of $p(\tau)$
requires further investigation, but the JADE data of $u_*$ show that $p(u_*)$ is reasonably Gaussian. Hence,
$$p(\tau) = \frac{1}{2\rho u_*} p(u_*), \qquad (7)$$

is skewed to smaller $\tau$, suggesting that the large-eddy model results of Klose et al. (2014) are qualitatively reasonable. Figure
8 shows that $u_*$ in Event-10 not only had a larger mean value but also a larger variance than in Event-11. We emphasize that
the variance of $u_*$ strongly affects saltation, because saltation flux depends non-linearly on $u_*$. To illustrate this, we consider
$u_{*1}$ and $u_{*2}$, and assume that
• $u_{*1}$ and $u_{*2}$ are Gaussian distributed and have the same mean that equals $u_{*t}$ (say 0.2 ms$^{-1}$)
• $u_{*1}$ and $u_{*2}$ have respectively standard deviation, $\sigma_1$ and $\sigma_2$, with $\sigma_2 = \eta \sigma_1$ and $\eta > 1$; and
• $Q$ satisfies the Owen's model (Owen, 1964),
$$Q_i = c u_{*i}^3 \left(1 - \frac{u_{*t}^2}{u_{*i}^2}\right) \qquad \text{for } u_* > u_{*t};$$

$$\text{otherwise } 0; \quad \text{with } i = 1, 2, \qquad (8)$$

where $c$ is a dimensional constant. It follows that the ratio of the mean values of $Q_2$ and $Q_1$ is
$$\eta_Q = \frac{\bar{Q}_2}{\bar{Q}_2} = \int_{u_{*t}}^{\infty} Q_2\, p(u_{*2})du_{*2} \Big/ \int_{u_{*t}}^{\infty} Q_1\, p(u_{*1})du_{*1}, \qquad (9)$$

Equation (9) can be evaluated numerically for different $\eta$ (Table 1) and is approximately
$$\eta_Q = 0.607\, \eta^2 - 0.0028\eta + 0.4283, \qquad (10)$$

This shows that $p(u_*)$ profoundly influences the magnitude of $Q$. For fixed $\bar{u}_*$, a larger $u_*$ variance corresponds to a larger $\bar{Q}$.
Table1. Streamwise saltation flux ratios, $\eta_Q$, for different $u_*$ std ratios, $\eta$ (see text for details).

| $\eta$ | 1.2 | 1.4 | 1.6 | 1.8 | 2 | 3 | 4 |
|---|---|---|---|---|---|---|---|
| $\eta_Q$ | 1.30 | 1.63 | 2.00 | 2.41 | 2.86 | 5.83 | 10.15 |

Second, in unstable conditions, turbulence is stronger due to buoyancy production, which leads to increased saltation
bombardment intensity. We do not have independent evidence to verify this, but to illustrate the point, we use a two-
dimensional (2-d, $x_1$ in mean wind direction and $x_3 \equiv z$ in vertical direction) saltation model (Supplement A) to simulate the
impact kinetic energy of saltation sand grains. For given $u_*$ and roughness length, $z_0$, a 2-d turbulent flow is generated with
the mean wind assumed to be logarithmic $\kappa \overline{u_1} = \overline{u_*} \ln(z/z_0)$ and the velocity standard deviations satisfy
$$\frac{\sigma_{u1}}{\overline{u_*}} = a \cdot ln\left(\frac{z}{z_0}\right), \tag{11}$$

$$\frac{\sigma_{u3}}{\overline{u_*}} = f_{u3}(\zeta) \cdot a \cdot ln\left(\frac{z}{z_0}\right), \tag{12}$$

and the dissipation rate for turbulent kinetic energy, $\varepsilon$, satisfies
$$\varepsilon \frac{\kappa z}{\overline{u_*}^3} = f_\varepsilon(\zeta) , \tag{13}$$

The similarity relationships $f_{u3}(\zeta)$ and $f_\varepsilon(\zeta)$ follow Kaimal and Finnigan (p16, 1995). As saltation takes place in the layer
close to the surface, the vertical profiles of $\sigma_{u1}$ and $\sigma_{u3}$ are considered following Yahaya et al. (2003). The coefficient $a$
(=1.16$\beta$) is varied by setting $\beta$ to 0.75, 1.00 and 1.25 for weak, normal and strong turbulence, respectively.
In each numerical experiment, 20000 sand grains of identical size are released from the surface and their trajectories are
computed. At impact on the surface, the particles rebound with a probability of 0.95 and a rebounding kinetic energy, $K_{reb}$,
0.5 times the impact kinetic energy, $K_{imp}$. The rebound angle is Gauss distributed with a mean of $40^o$ and standard deviation
$5^o$. Splash entrainment is neglected. The PDF of $K_{imp}$, $p(K_{imp})$, is used as a measure for bombardment intensity.
Many numerical experiments were carried out, but for our purpose, we show only the results of the ones listed in Table 2.
The initial velocity components of sand grains ($V_{1o}$, $V_{3o}$) are generated stochastically. $V_{1o}$ is Gauss distributed with a mean
$\bar{V}_{1o} = \bar{u}_* \cos(55^o)$ and standard deviation, $\sigma_{V1o} = 0.1\bar{u}_*$. $V_{3o}$ is Weibull distributed with a shape parameter $A = 2$ and a scale
parameter $B' = \bar{u}_* \sin(55^o)/\Gamma(1 + 1/A)$ where $\Gamma$ is a Gamma function. To account for the influence of stability on $V_{3o}$, $B'$
is modified such that the adjustment to $\sigma_{V3o}$ is the same as that to $\sigma_{u3}(10z_0)$, i.e., the modified scale parameter, $B$, is given
by
$$B = \beta f_{u3}\left(\frac{10z_0}{L}\right)B'. \tag{14}$$


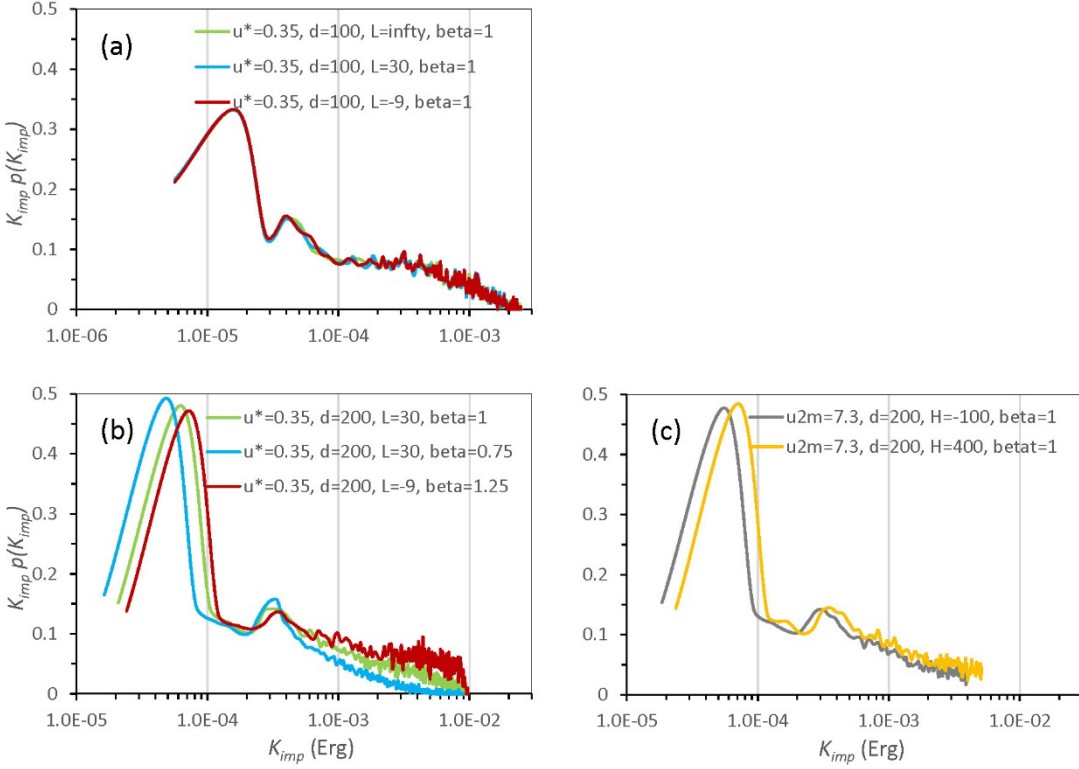


Figure 9. Probability density function $p(K_{imp})$ (plotted in $K_{imp}\,p(K_{imp})$ against $K_{imp}$ in logarithmic scale) for the numerical experiments. In
(a), $p(K_{imp})$ is shown for $u_* = 0.35\text{ms}^{-1}$, $d = 100\mu m$ and $\beta = 1$ but for three different Obukhov lengths $L = \infty$, 30m and -9m. In (b), the effect
of $\beta$ on $p(K_{imp})$ is examined; and in (c) the effect of stability on $p(K_{imp})$ with given mean wind speed at $z = 2m$ is examined.
Figure 9a compares $p(K_{imp})$ for Exp1a, 1b and 1c and shows that $p(K_{imp})$ for these cases is very similar. The small
differences in $p(K_{imp})$ between the cases suggest that the differences in particle trajectory arising from the stability
modification to turbulence profile, with $u_*$ fixed, are negligible. However, a small change in $\beta$, as Figure 9b shows for Exp2a,
2b and 2c, can lead to significant changes in $p(K_{imp})$ with larger $\beta$ corresponding to higher probability of larger $K_{imp}$, namely,
high saltation bombardment intensity. In Exp3a and 3b, $u_{2m}$ (mean wind 2m height) is set to 7.3 ms$^{-1}$ and the surface sensible
heat flux, $H$, to -100 and 400 Wm$^{-2}$. Figure 9c shows that $p(K_{imp})$ differs with larger $K_{imp}$ in unstable conditions.
Table 2: Numerical experiments for saltation bombardment intensity. For all experiments, $z_0 = 0.48mm$, $C_0 = 5$, $C_1 = 2$ and $\rho_p = 2650$ kgm$^{-3}$.

| Exp | $u_*$ (ms$^{-1}$) | $L$ (m) | $d$ (µm) | $\beta$ |
|---|---|---|---|---|
| Exp1a, 1b, 1c | 0.35 | $\infty$, 30, -9 | 100 | 1.0 |
| Exp2a, 2b | 0.35 | 30 | 200 | 0.75, 1 |
| Exp2c | 0.35 | -9 | 200 | 1.25 |
| Exp3a, 3b | $u_{2m}$=7.3 | $H$=-100; 400 Wm$^{-2}$ | 200 | 1 |


To summarize, the numerical experiments suggest that the PDF of the particle initial velocity significantly influences the
saltation bombardment intensity, and saltating particles in unstable ABL impact the surface with larger kinetic energy than in
stable ABL. This is the result seen in Figure 7 and 8, i.e., saltation in Event-10 was more fully developed than in Event-11.
The more fully developed saltation in unstable ABL increases saltation bombardment intensity and hence the release of finer
dust particles, seen in Figure 6.
**4.2 Influence of Surface Condition on Dust PSD**
A detailed analysis of Event-12 (Figure 10) reveals that the dependency of dust PSD on friction velocity and ABL stability
is made complicated by soil surface conditions. To analysis how dust PSD evolved during the event, we divide Event-12
which lasted ~5.5 hours, into 11 half-hourly time sections labelled as S1, S2 etc. For each section, the dust PSD is averaged
over time and plotted in Figure 10c. Figure 10a shows the time series of $Q$, $w_*$ and $u_*$, and Figure 11b those of 2 cm soil
temperature and soil moisture. For the whole event, the ABL was unstable, with $w_*$ fluctuating around $1.64 \pm 0.12$ ms$^{-1}$.
Initially (e.g. S1 and S2), $u_*$ was relatively large, exceeding 0.4 ms$^{-1}$ at times, but then eased to around 0.3 ms$^{-1}$. $Q$ generally
followed the variations of $u_*$. Yet, the dust PSD showed a systematic shift from coarser to finer particles, as the event
progressed. The dust PSD dependency on $u_*$ of Event-12 does not conform with the results for Event-11 (Figure 6) and the
overall results (Figure 4a). Ishizuka et al. (2008) noticed that prior to Event-12, weak rainfall occurred (R4, Figure 5b) and
consequently, weak crusts formed on the soil surface. Apparently, the lightly crusted surface prevented the emission of fine
dust particles in the early stages of Event-12. As the event progressed, soil temperature increased, soil moisture decreased
(Figure 10b) and the saltation during the early stages caused the destruction of the crusts and the amount of fine dust
particles available for emission increased. These are the most likely reasons for why in the later stages of Event-12, an
increased fraction of fine dust was released, although the atmospheric stability did not significantly change and $u_*$ actually
decreased.

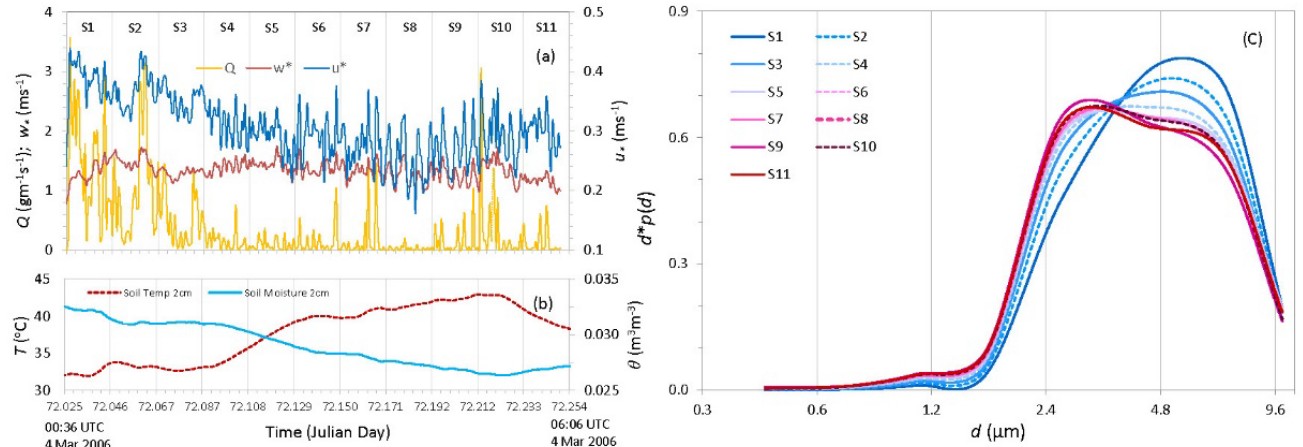


Figure 10. (a) Time series of streamwise saltation flux, $Q$ (gm$^{-1}$s$^{-1}$), convective scaling velocity, $w_*$ (ms$^{-1}$), and friction velocity, $u_*$ (ms$^{-1}$),
for Event-12. The time span of Event-12 is divided into 11 half-hourly sections, labelled as S1, S2 etc. (b) As (a), but for soil temperature,
$T$ (°C), and soil moisture, $\theta$ (m$^3$m$^{-3}$), both at 0.02m depth. (c) Dust PSDs averaged over section S1, S2 etc.

### 4.3 Uncertainties

Several issues are related to the uncertainties of the analysis. First, the approximation of emission-dust PSD with airborne-
dust PSD measured at some height above ground causes uncertainties, because airborne-dust PSD is height dependent as
consequence of the dust-transport processes (e.g. diffusion and deposition) in the atmosphere, which are both particle-size
and turbulence-property dependent. As our understanding of these processes is not complete and dust measurements have
inaccuracies, a careful selection of the data for the analysis is necessary. Figure 11 shows a comparison of Event-10
averaged airborne-dust PSDs at 1 m and 3.5 m. Ishizuka et al. (2014) suggested to exclude the 2m-OPC data, because they
do not correlate well with the 1 m- and 3.5 m-OPC data. The PSDs derived from the 2m-OPC data do show unexpected
differences in comparison to those from the 1m- and 3.5m-OPC data. We thus have excluded the 2m-OPC data from our
analysis. The PSDs derived from the 1m- and 3.5m-OPC data somewhat differ, with the peak particle size shifted by about
two microns, i.e., airborne-dust PSD has a noticeable change with height. This also implies that it would be very difficult to
compare airborne-dust PSD measured at different locations and under different conditions without a well-established
framework equivalent to the Monin-Obukhov similarity theory.
Also shown in Figure 11 is the Event-10 averaged emission-flux PSD calculated using Equation (3a). Dust fluxes for
different particle size bins are calculated using the 3.5m- and 1m-OPC data with the gradient method (Gillette et al. 1972)
and corrections (Shao et al. 2011). As dust flux is proportional to the negative gradient of dust concentration, emission-flux
PSD basically describes how dust-concentration gradient [in our case $-(c_{3.5m} - c_{1m})$ ] depends on particle size.

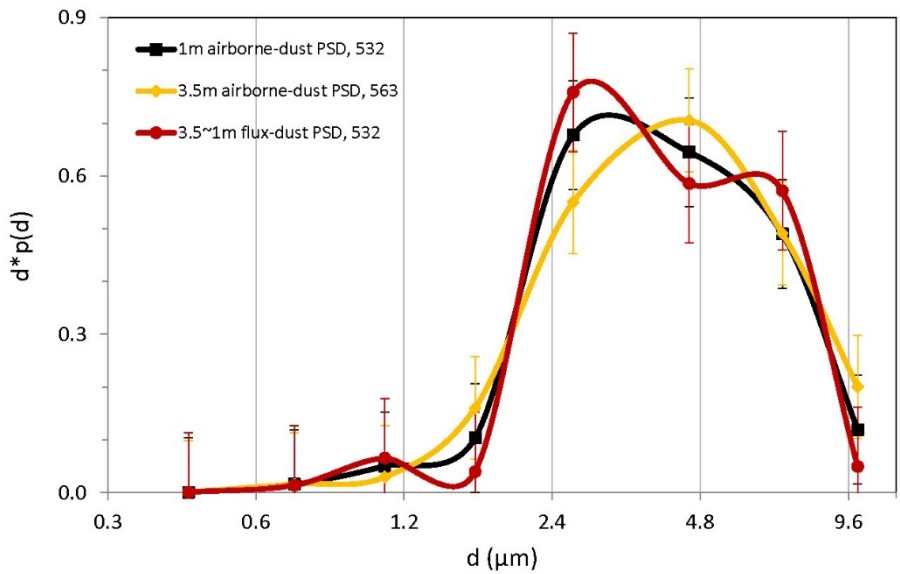


Figure 11: JADE Event-10 averaged airborne-dust PSD measured at 1m (532 one-minute samples) and 3.5m (563 one-minute samples) using OPCs. Also shown are standard-error bars. For comparison, Event-10 averaged (over 532 one-minute samples) emission-flux PSDs calculated using Equation (3a) is plotted.

Although dust PSDs derived from 1m-OPC and 3.5m-OPC data differ, qualitatively they show similar dependencies of dust PSD on $u_*$ and $w_*$. Figure 12a compares the averaged dust PSDs for two $u_*$ categories using the 1m-OPC and 3.5m-OPC data. For both cases, the dust PSD dependency on $u_*$ is visible. Figure 12b compares the averaged dust PSD for a given $u_*$ category (0.35 ~ 0.45 ms$^{-1}$) under stable and unstable conditions. Again, both the 1m-OPC and 3.5m-OPC dust PSDs show dependency on $w_*$.

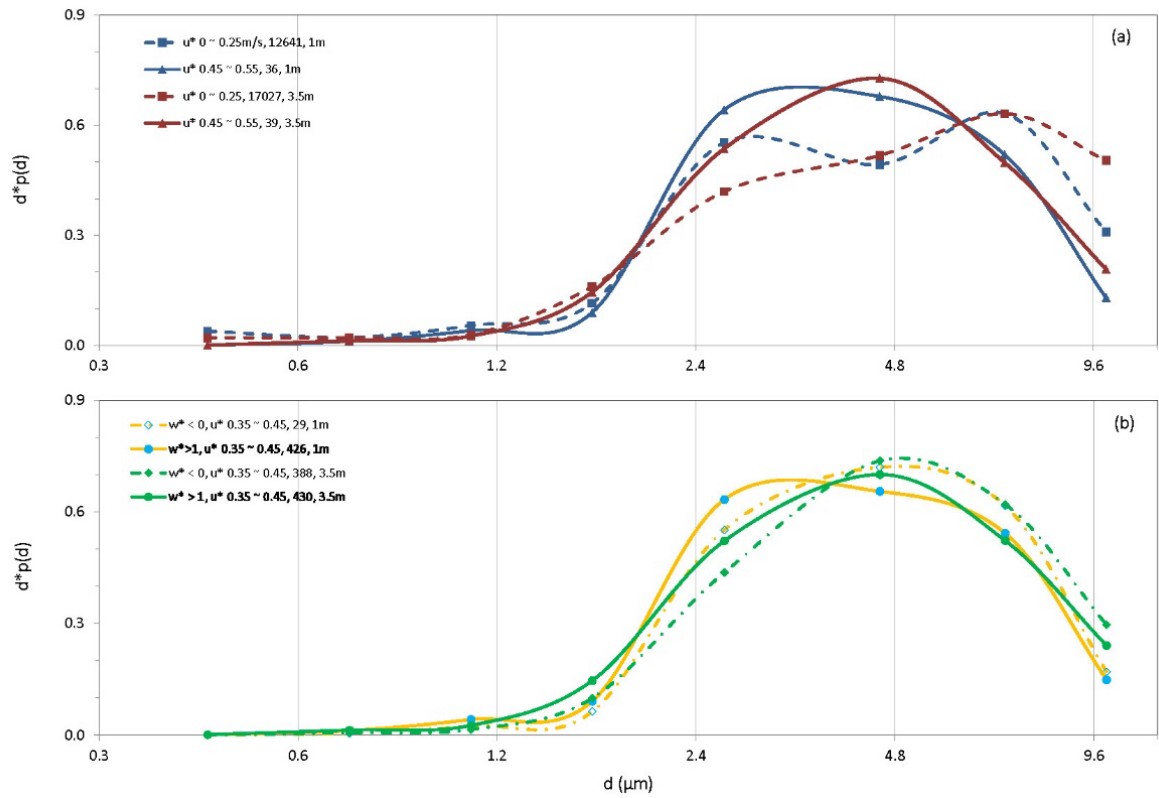


Figure 12: (a) JADE averaged airborne-dust PSD measured at 1 and 3.5m for two $u_*$ categories. (b) As (a) but for one $u_*$
category and two different stabilities.

It needs to be clarified whether using 1-minute averages of shear stress, saltation flux and dust flux are appropriate for the
study. Related to this question are two inter-wined yet somewhat different scaling issues, namely, (1) the scaling of turbulent
flux and the corresponding mean variable of boundary-layer turbulent flows (i.e. the flux-gradient relationship); and (2) the
scaling of aeolian fluxes and atmospheric forcing (i.e. saltation/dust-emission intermittency). It is usual in boundary-layer
meteorology to compute a turbulent flux from the profile of the corresponding mean quantity, e.g., mean shear stress from
mean wind profile, and the time interval for the mean is typically 15 to 30 minutes such that the assumptions of horizontal
homogeneity and stationarity commonly made in boundary-layer studies are met. This issue is not yet fully resolved even in
boundary-layer studies. For example, large-eddy models (with spatial resolution of several meters and temporal resolution of
seconds) frequently use the Monin-Obukhov similarity functions to estimate sub-grid surface stress from the grid-resolved
speed. In this study, we distinguish the 1-min averages of $u_*$ from the mean shear stress of the boundary-layer flow to
emphasize the importance of shear stress fluctuations. The problem how to scale aeolian fluxes is not new (e.g. Shao and
Mikami, 2005). Dupont (2020) has a dedicated paper on this problem and stated that $u_*$ is a suitable scaling parameter for
dust flux over usual 15~30-min time intervals, but at smaller time resolution, wind becomes more relevant to scale dust
fluxes, a conclusion similar to that reached in Sterk et al. (1998). The studies of Stout and Zobeck (1997) and Sterk et al.
(1998), and more recently Klose and Shao (2012) and Klose et al. (2014), all pointed to the importance of taking
instantaneous shear stress into consideration of aeolian dynamics. As Shao (2008, p203-205) explains, $\tau_{inst}$ is proportional to
$U'^2$, where $\tau_{inst}$ is instantaneous shear stress and $U'$ instantaneous wind speed. The argument of Shao (2008) reasonably well
explains the conclusions of Sterk et al. (1998) and Dupont (2020). Liu et al. (2018, Figure 7) analysed co-spectrum of
saltation flux and shear stress and showed that they have a correlation peak at $2 \times 10^{-3}$Hz, corresponding to gusts/large eddies
of around 10 minutes in turbulent flows. These considerations suggest that to average shear stress and aeolian fluxes over
one minute is appropriate and has the advantage of showing how dust emission is related to turbulence. We have emphasised
throughout this paper that turbulence is key to understanding the dependency of dust PSD on ABL stability, because the
most essential difference among ABLs of different stability are the intensity and structure of turbulence.
As far as averaged dust PSDs are concerned, we have compared the dust PSDs averaged for different $u_*$ categories using
1-minute averaged data and 10-minute averaged data. The results are almost the same.

## 5  Conclusions

Using JADE data, we showed that dust PSD is dependent on friction velocity $u_*$. This finding is consistent with the wind-
tunnel study of Alfaro et al. (1997). The JADE data support the claim that dust PSD is saltation-bombardment dependent and
does not support the hypothesis that dust PSD is invariant.
The JADE data show that dust PSD, as well as saltation PSD, also depends on ABL stability. This finding is consistent
with the results of Khalfallah et al. (2020). Dust PSD is dependent on ABL stability for two reasons. First, $u_*$ is a stochastic
variable and the PDF of $u_*$ profoundly influences the magnitude of saltation flux, $Q$, because of the non-linear relationship
between $Q$ and $u_*$. With fixed $u_*$ mean, a larger $u_*$ variance corresponds to a larger $Q$. Unstable ABL has in general larger $u_*$
variances which generate stronger saltation bombardment and produce the emission of finer dust particles. Second, in
unstable ABL, turbulence is generally stronger and in strong turbulent flows, the proportion of saltation particles with large
impacting kinetic energy is larger than in weak turbulent flows. Consequently, saltation in unstable ABLs is more fully
developed and saltation bombardment has higher intensity.
The dependencies of dust PSD on $u_*$ and ABL stability are ultimately attributed to the statistic behaviour of $u_*$, i.e., its
PDF $p(u_*)$, or more simply its mean and variance. These dependencies point to the same fact that, for a given soil, saltation
bombardment plays a determining role for the dust PSD. Stronger saltation causes in general the emission of finer dust.
The dependency of dust PSD on $u_*$ and ABL stability is made complicated by soil surface condition. In the case of strong
saltation and very weak surface/particle binding, the dust PSD dependency on $u_*$ may become less obvious. In the case of
strong surface/particle binding, dust emission in certain size ranges may be prohibited.

*Data availability*. Data can be accessed by contacting the corresponding authors.

*Author contributions.* Yaping Shao performed the data analyses and drafted the manuscript. Jie Zhang and Ning Huang contributed to the conception of the study, the data analysis and the writing of the manuscript. Masahide Ishizuka, Masao Mikami and John Leys conceived, designed and performed the experiments and helped finalize the paper.

*Competing interests.* The authors declare that they have no conflict of interest.

*Acknowledgments.* We thank the National Key Research and Development Program of China (2016YFC0500901), the National Natural Foundation of China (11602100, 11172118) and the Fundamental Research Funds for the Central Universities (lzujbky-2020-cd06) for support. The JADE project was supported by Kakenhi, Grants-in-Aid for Scientific Researches (A) from the Japan Society for the Promotion of Science (Nos. 17201008 and 20244078) and the Lower Murray-Darling Catchment Management Authority. We are grateful to the three referees for their constructive comments and to Dr. Sylvain Dupont and Dr. Jasper Kok for helpful discussions.

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
