# Peer review of "Dependency of Particle Size Distribution at Dust Emission on Friction Velocity and Atmospheric Boundary-Layer Stability"

_Atmospheric Chemistry and Physics, 2020_

## Referee Comment (RC1) · Anonymous Referee #1 · 26 May 2020

Review of 'Dependency of Particle Size Distribution at Dust Emission on Friction Velocity and Atmospheric Boundary-Layer Stability' by Y. Shao et al.

General comments: The size distribution (emission PSD) of the dust particles emitted from arid and semi-arid surfaces is crucial for the assessment of their numerous environmental impacts. However, this question has been much debated in the last two decades. Some experimental and theoretical studies emphasized the importance of the kinetic energy (increasing with the speed of the wind) of the saltating sand grains for this PSD, whereas others maintained that the PSD was constant. In addition, a quite recent study also showed that the stability of the atmosphere played a role: the

more unstable the lower atmosphere, the richer the PSD in fine particles. The aim of the present study is to re-analyze the data of the JADE experiment carried out in Australia for answering the following questions: 1) does the emission PSD depend on wind speed or not?, 2) is it confirmed that the atmospheric stability plays a part?, and 3) why? For answering these questions, the authors select two contrasted events of the JADE field campaign: one that occurred during a hot day (Event 10) with unstable conditions, and another (Event 11) that happened immediately after but in stable nocturnal conditions. Then, they analyze separately the influence of 1) u\* (and its probability distribution), and w\* (a proxy of atmospheric stability) on the emission PSD. My opinion is that the demonstration of the authors is guite convincing, well organized, and their methods are scientifically sound. Finally, the study brings final answers to ancient questions: yes, the emission PSD does depend on wind speed (through the kinetic energy of the saltators), and yes atmospheric instability promotes the ejection of finer particles (because of the widening og the u\* distribution as turbulence increases). In summary, this paper is a very important one and undoubtedly deserve publication in ACP after the few following comments have been addressed:

Comments and minor concerns: 1) P. 1, line 26: F increases with Q, but it is not simply proportional to it. For a given surface, the so-called 'sandblasting-efficiency' (ratio of F to Q) is usually found to vary with u\*. 2) P. 2, lines 45 and 48; correct the years for Kok's publications 3) P. 2, line 62: in Ishizuka et al. (2008) and in this paper, u\* is calculated over periods of 1'. This is too short to integrate the time-scales of turbulence in the surface layer (Dupont el al., 2019), but presents the advantage of following more closely the variations of the instantaneous wind speed to which saltation responds in quasi real-time. Conversely, Khalfallah et al. (2020) calculated u\* over longer periods of 16'. The smoothening of the u\* statistical distribution resulting from this averaging probably explains that they could not detect any notable influence of u\* on the emission PSD. 4) P.3 line 64: JADE (not JADA) 5) P.4, line 88: In sedimentology, the texture of a soil is defined from the size-distribution of its particles after full dispersion. So, the soil is sandy loam. 6) P.6: Figure 4a. In the insert, the same color code as in the
rest of the figure could be used to differentiate Event 10 from Event 11. 7) P.6; line 144: 'do not substantially differ'. What does this mean? 8) P.8; line 177: what are the implications of the fact that the distribution of ïĄt' is skewed to smaller values? 9) P. 10, line 208: experiments 'were'... 10) P. 113, lines 229-235: Please rewrite this part to avoid repetitions and clumsy formulations.

---

## Short Comment (SC1) · 8 Jun 2020

I had great interest reading this paper. It is well written and well structured. The question of the sensitivity of the particle size distribution (PSD) of emitted dust on the friction velocity  $(u^*)$  and thermal stability is indeed crucial and of great interest for the erosion community. In that sense, this study tries to answer to this question using the JADE field experiment.

This study follows the recent paper of Khalfallah et al. (2020) that showed a dependency of the emitted dust flux PSD to the atmosphere thermal stability using the WIND-O-V field experiment, for stability conditions ranging from near-neutral to slightly unstaPrinter-friendly version

ble (never stable as here). This was attributed by these authors to the dependency of the particle eddy diffusivity to the particle size. To be honest, my own analysis of the WIND-O-V data did not lead me to the same conclusions. I did not find any dependence of the emitted dust flux PSD to the atmosphere thermal stability. The enrichment in fine particles of the dust flux with increasing unstability claimed by Khalfallah et al. is in fact an impoverishment of their flux in coarse particles. In their statistics, this impoverishment and thus the stability dependency of the dust flux PSD, resulted only from few periods in two of their eight events, periods that should have been discarded due to a too low difference in dust concentration between their two dust concentration levels, not permitting the applicability of the flux-gradient approach. I am also skeptical about their justification of the PSD stability dependency based on the variability of the particle size (0.3 to 9  $\mu$ m) since particle trajectory-crossing effect should be quite negligible for such small particles. I would therefore not rely much on Khalfallah et al. (2020).

Interestingly, the present paper concludes as well on a larger fraction of fine particles of the emitted dust PSD in unstable conditions (event 10), by comparison with a stable condition event (event 11). As opposed to the particle eddy diffusivity argument of Khalfallah et al. (2020), this difference of dust PSD with stability is explained here by difference in saltation-bombardment intensity with stability. In particular, the saltation PSD was found different between the unstable and the stable events for similar u\* values.

While I am convinced by the sensitivity of the dust PSD to u\*, I am less convinced that there is also a direct sensitivity to the atmosphere thermal stability. In my opinion, there is not other thermal stability effect than the one already observed in u\* (Eq. 5). You argued that the larger turbulence in unstable conditions enhances saltationbombardment intensity, and thus emission of finer dust particles, as compared to stable conditions with identical u\*. In my opinion, your argument that the flow turbulence intensity is larger in unstable than in stable conditions for identical u\* is not demon-
strated and disagrees with Fig. 1.9 of Kaimal and Finnigan (1994, p20). I therefore think that the differences observed on the saltation and dust PSD between unstable and stable cases (daytime and nighttime conditions, respectively) for similar u\* are explained by something else than thermal stability. To the end, you just observed a difference of PSD of near-surface dust concentration and saltation between only one daytime event and one nighttime event for similar u\*, and you simply attributed this difference to thermal stability but without any convincing demonstration. This could be explained by many other reasons. This demonstration of the thermal stability effect is important as I am afraid that another paper suggesting a thermal stability effect on dust PSD in continuation of the erroneous paper of Khalfallah et al. (2020) would leads the erosion community in a wrong direction. This explains this comment.

In my opinion, more figures should be presented in order to better characterize the events and to be more convincing:

- In order to understand what happened during the erosion events 10 and 11, I would expect additional figures showing in particular the time variation of the dust and saltation PSD during the events. For example, a 2D plot of the time variation of the dust and saltation flux PSD would be helpful. The figures presented in the manuscript only show averages over the whole events while both events are very different in time scale (10h for the event 10 and 6h for event 11) and more importantly in stationarity. The mean wind speed is relatively stationary during event 10 and very unstationary during event 11 (Figure 2b). The event 10 covers stability from near-neutral to unstable conditions; we would therefore expect to see as well a sensitivity with time of the dust PSD related to the stability variation for a similar u\* if this dependency exists.

- The PSD of the emitted dust flux should be also presented to characterize the PSD of emitted dust and not only the PSD of the airborne dust measured here at 1, 2 and 3.5 m. A background concentration, in particular of fine particles, could be present during the events without any relation with dust emission, and the depth of the atmospheric layer where dust particles are dispersed should be quite different between the unstable

**ACPD**
and stable cases as well as along the daytime event 10. Showing the difference of dust flux PSD between events 10 and 11 and their time variation would be much more convincing.

You should discuss other possible reasons (or discard them) for the difference of the dust and saltation PSD between the events 10 and 11:

- The event 11 occurred at night. Could it be possible that droplet condensation before or between the intermittent sub-events reinforced inter-particle cohesion and thus explains the lower fraction of fine dust particles of the emitted dust PSD at night?

- Event 11 also occurred after a strong sand drift. Could this strong sand drift have changed the PSD of mobile (weakly attached) particles at the surface, explaining the larger proportion of 70-80  $\mu$ m particles in the saltation PSD for small u\* (beginning of the event)? A time variation of saltation PSD would be helpful.

In the last part of the paper, the three justifications (called perspective in the text, lines 159-160) for explaining the dependency of the dust PSD on  $w^*$  (stability) based on the saltation bombardment intensity, should be much more convincing:

- The first justification is related to the ABL similarity theory, showing that u\* depends on the thermal stability (Eq. 5). In my opinion, this justification is not relevant here. It just shows that stability is already accounted for in u\* and thus it does not demonstrate a direct relation between dust PSD and thermal stability for a constant u\*.

- The second justification is on the stochasticity of u\*, whose variance changes with stability. By definition, u\* is a mean quantity characterizing indirectly the amount of momentum absorbed by the surface (square root of the absolute value of the momentum flux), by accounting for all eddy scales transporting momentum. To respect this last point, u\* is usually estimated over 15 to 30 min time period for surface atmospheric boundary layer turbulence (Dupont et al. 2018). For an ideal event with stationary large-scale wind conditions and constant thermal stability (as in wind-tunnel), u\* should

**ACPD**
be constant in time with no variance whatever the thermal stability. I therefore do not understand this argument or justification. The stochasticity of u\* that you are observing in Figure 6 is due (1) to the non-stationarity of the mesoscale wind during the erosion events (see Figure 2b), and (2) to your choice of estimating u\* at only 1 min time scale. Furthermore, eddies transporting momentum are smaller in stable condition than in unstable conditions. Your choice of using 1 min to estimate u\* is questionable, especially for the event 10, as it certainly misses large-scale contribution to the momentum transport (see Dupont 2020), and the time period for estimating u\* should be certainly smaller for stable (event 11) than unstable (event 10) conditions. The variance of u\* in figure 6 depends mainly on the stationarity of the wind during the events and your choice of estimating u\* at 1 min, but much less on the thermal stability. A more stationary large-scale wind condition during event 10 with the same thermal stratification would have induced a smaller variance of 15-min u\* than during event 11. I am also not sure that the u\* PDF of event 11 reflects the variability of u\* during an erosive event in stable condition, as this PDF seems dominated by several periods without erosion following Figure 2a. For this reason, I am not convinced neither by this justification of a direct dependence of the dust PSD on the thermal stability. As I wrote above, this stochasticity of u\* is only related to large-scale variations of the wind and to your choice of computing u\* at 1 min, which leads to not comparable u\* between unstable and stable conditions.

- The third justification is on the enhancement of the saltation bombardment intensity with buoyancy production of turbulence. To demonstrate this point, a saltation model is used. The idea is to demonstrate that the flow turbulence intensity increases, and so the saltation bombardment intensity, with unstability while keeping u\* constant. This means that additional turbulence not transporting momentum is produced with increasing unstability, for stability conditions that are still close to neutrality since the wind speed remains quite significant during erosion events. The model demonstrates that for a similar u\*, the thermal stability represented here by L (Obukhov length) does not modify the impaction energy of saltating particles (figure 7a). This is, therefore, a strong

**ACPD**
indication that stability does not directly impact saltation PSD and does not increase turbulence for identical u\*. By only artificially increasing the turbulence of the flow while keeping u\* constant (and with no variance, i.e. no stochasticity, in contradiction with the second justification), obviously the impaction energy of saltating particles is increased (figure 7b). Here, it was assumed that this additional turbulence is not efficient at transporting momentum (and so increasing u\*). This artificial increase of turbulence has, however, no justifications. Consequently, this cannot constitute a demonstration. The last simulated case shows that for similar wind speeds, the impaction energy of saltating particles is increased with increasing unstability (sensible heat flux, figure 7c) but this is certainly not for a constant u\*! Here, u\* should be smaller for the negative heat flux case (low turbulence) than for the positive heat flux case. To the end, this third justification does not demonstrate as well a direct dependence between the saltation PSD (and this dust PSD) and the thermal stability.

In my opinion, near the surface where saltation occurs the turbulence is largely dominated by friction and much less by buoyancy. During erosion events, we are far from free convection conditions because of the strong wind, even during daytime. I doubt that an additional turbulence not transporting momentum would be significant in event 10 as compared to event 11. Following Kaimal and Finnigan (1994, Fig. 1.9, p20), the cross-correlation coefficient for the momentum flux (ruw), which gives an information on the ratio of the momentum flux compared to the level of the flow turbulence, is almost constant for -1 < z/L < 1 in the surface layer, i.e. for stability conditions where the Obukhov length is larger than 1 m, which should largely include events 10 and 11. This figure of Kaimal and Finnigan represents a strong evidence that the argument suggested in this paper for the dependence of the dust PSD on thermal stability based on the increased turbulence of the flow with unstability while keeping constant u\*, is doubtful.

Hope that my comment is helpful and constructive.

Sincerely
**Sylvain Dupont**

References:

Dupont, S. (2020). Scaling of dust flux with friction velocity: time resolution effects. Journal of Geophysical Research: Atmospheres, 125, e2019JD031192. https://doi.org/10.1029/2019JD031192.

Dupont, S., Rajot, J.-L., Labiadh, M., Bergametti, G., Alfaro, S. C., Bouet, C., Fernandes, R., Khalfallah, Lamaud, E., Marticorena, Bonnefond, J.-M., Chevaillier, Garrigou, D., Henry-des-Tureaux, T., Sekrafi, S., Zapf (2018). Aerodynamic parameters over an eroding bare surface: reconciliation of the law of the wall and eddy covariance determinations. Journal of Geophysical Research - Atmospheres, 123 (9), 4490-4508.

Kaimal, J. C., & Finnigan, J. J. (1994). Atmospheric boundary layer flows. Their structure and measurements. New-York: Oxford University Press.

---

## Referee Comment (RC2) · Anonymous Referee #2 · 15 Jun 2020

This paper addresses two key-questions that are debated by the scientific community working on wind erosion and atmospheric dust. The first question, debated from several years, concerns the dependence or not of the emission-dust PSD with the wind friction velocity. The second one, more recently laid on the table by Khalfallah et al. (2020), deals with the dependence of the same emission-dust PSD with the atmospheric boundary-layer stability. These questions are under debate mainly because obtaining relevant observations in natural conditions to investigate such dependencies is difficult. Long and complex campaigns are required for this, a drastic selection among the observations is necessary to isolate the best situations and, therefore, relevant data are scarce. Thus, one of the first interest of this paper is to propose a

re-analyze of the data from the JADE campaign (Ishizuka et al., 2008). Beyond that, this article offers an original and in-depth analysis of these data. Finally, it proposes numerical simulations to support the conclusions. More precisely, the paper provide convincing observations on the dependency of emission-dust PSD with the wind friction velocity and atmospheric boundary-layer stability. The experimental results reported in Figures 4, 5 and 6 are the key figures of this paper. The discussion on the PDF of u* and on its role on the intensity of the saltation is really convincing. The authors also propose interesting and elegant explanations for these dependencies by examining the role played by the wind friction velocity PDFs in cases of high and low u* and of different stability conditions. I am not sure that this paper will definitively close the debate (more experimental data will probably be necessary for this) but I am sure that this document is an important contribution in order to better understand what controls PSD dust emissions and to elaborate future experiences. The paper is well written and structured even if additional information is needed in some places. I recommend to publish this paper with minor corrections. 1. Introduction: The introduction is very well written, concise and gives a clear idea of the scientific context. lines 41-42 (and also mentioned on line 144-145): I agree with the authors that dust-airborne PSD measured very close to the surface can probably be assimilated to dust-emission PSD by assuming that the difference of the particle diffusivity compared at that of other scalar (and thus, its dependency on particle diameter) can be neglected. However, because this assumption is an important point of the paper, it deserves to be better discussed, especially because there are few experimental data on particle diffusivity, especially for small particles, and that most of the information we have come from models. In the same way, since the authors indicate that size-resolved dust fluxes were measured during JADE (line 67) a comparison between dust–airborne PSD and dust-flux PSD should be added to support this assumption, at least in the supplement. line 49: the only reference to the Pisso et al., 2019 's paper is not sufficient to support the statement that "The proposed emission-dust PSD is frequently used in dust models". line 52: replace airborne-dust PSD by "dust flux PSD" or "emission dust PSD"

2. JADE data line 64: replace JADA by JADE I know that the JADE experiment has been described in details in various papers, especially in Ishizuka et al., 2008, and does not need to be described again. However, the presentation of the data in this paper is too short: for example, it is never indicated that OPCs measured number concentrations that are further converted into mass concentrations assuming spherical particles of density= 2380 kg m-3. In the same way, Ishizuka et al. (2014) mentioned that the measurements for the bin 0.3-0.6 $\mu$m were not considered because of a significant difference between the two OPCs for this bin. In the present paper, this bin is used and the reason for that should be explained, at least to be consistent with the previous paper. However, it must be noted that this point is not critical for this paper since the contribution of this bin is negligible in mass as shown on figures 3 and 4. In the same way, the data from OPC measuring at 2 m high were not considered as relevant in Ishizuka et al. (2014) (because they does not correlate with the OPC measurements performed at the other heights) but are used in this paper. This should also be explained. Finally, line 96, it is indicated that no rainfall occurred as a consequence of the cold front crossing but in Ishizuka et al; (2008), it is mentioned that the rain sensor detected several very small precipitation events that have been also observed by the authors. Even if, as mentioned in Ishizuka et al. (2008) the drying of the soil was very rapid, this should mentioned. Figure 1: The measured dust concentrations are very high (several mg m-3!) even for event 11. This should be underlined in the text. Indeed, since the authors used airborne-dust PSD and not dust flux PSD, it is necessary to provide arguments showing that the measured dust PSD can be directly linked to dust emission. And such high concentrations of dust strongly suggests that the contribution of advection to the measured dust PSD is probably very limited. line 88: reformulate. A soil has only one texture. A formulation such as "The results of the analyze by method A correspond to (or suggest) a loamy sand texture while the results of the analyze by method B…" should be better. line 91: I appreciate the method consisting to select among various dust events those the best adapted to the objective of the study. However, the justification for selecting these two events is very short ("Event-10

occurred under daytime unstable, while Event-11 under night-time stable, conditions"). I imagine that they were other events among the 12 aeolian events recorded during JADE that occurred also in stable or unstable conditions. Are these events selected because the stability conditions were particularly constant for these two events? 3. Results

Figure 3: we have no idea of the number of points contributing to each u* category. This should be added in the legend of the figure; in the same way, no information is provided on the spread of the different points that allow to construct the PSD (standard deviation bars should be added). The authors write "that dust PSDs for Event-10 and -11 considerably differ": maybe an additional panel reporting the difference between PSD10 and PSD11 for similar u* categories could better illustrate these differences in PSD.

Figure 4: This figure in which are averaged all heights and all u* is important since it clearly shows that the dust and saltation PSD shift between the two events. However, this averaging approach is not well introduced and it should not be obvious for the reader to understand why it is useful and relevant to make such averaging. The paper should explain that. The insert is too small and should be a figure by itself. Same comment concerning standard deviations as for figure 3.

On figure 6, there is a shouldering in the high values for the observed u* distribution corresponding to event 11. This suggests that the u* PDF could be bimodal for this event. Moreover, the adjusted Gaussian PDF for u* does not include this shouldering reducing the variance of the u* Gaussian PDF for event 11. This should be discussed.

The numerical simulations are interesting and illustrate the sensitivity of the impact kinetic energy to different parameters on which the stability conditions could act. They clearly suggest that larger variances in u* PDF, as generally observed in unstable conditions, generate stronger saltation and thus should be responsible for higher production of fine particles.

---

## Referee Comment (RC3) · Anonymous Referee #3 · 25 Jun 2020

Overall assessment

I read the manuscript from the perspective of a specialist in wind erosion and dust emission but a generalist to this specific focus of the manuscript. The authors provide a clear, logical development of the focuses of their manuscript on the different bases for explaining particle size distributions of emitted dust and the dependency of airborne-dust PSD on atmospheric boundary layer (ABL) stability. The topic is valuable for the community and the work is well presented. However, I am not convinced by the approach used in the manuscript. I think the work in the manuscript omits uncertainty. If that uncertainty were included, I think it may lead to different / alternative conclusions.

[Figure]

Therefore, to increase confidence in the results I think the omitted uncertainty must be tackled, in some form or other, before the work can be published. I provide below additional information on this point. I also think that some improvements in the structure of the manuscript will help the reader more easily follow the explanation of the work.

In short, I am thoroughly supportive of the work. I think the manuscript needs to be revised to give confidence that the results are indeed detectable and therefore inter-pretations are robust to the uncertainty. The nature of the revisions I describe below I think, are consistent with a major revision, despite not being too difficult to achieve in a short period of time, if all other things were equal.

Main issues

Wind friction velocity uncertainty

In the abstract, it is stated that friction velocity $u*$ is a surrogate for surface shear stress and descriptor for saltation bombardment intensity. Line 32 states that "for a given soil, the particle size distribution of dust at emission (emission-dust PSD), $ps(d)$, must depend on saltation bombardment or on friction velocity". The JADE field measured data are used to show that the (finely resolved) particle size distribution is dependent on measured wind friction velocity.

In contrast to this approach, it is well known (cf. sediment transport models) that the wind energy available for saltation bombardment is not $u*$, it is the energy $us*$, which remains after wind momentum has been extracted by the roughness 'canopy'. In other words, $us*=u*.R$ where $R$ is the partition of drag. Under controlled conditions with homogeneous material, smooth surface (bed) without ripples and the bed reset after each experiment, it may be reasonable to assume that $R=1$. However, the authors use field data which, even under the smoothest field conditions are very likely to cause $R$ not equal to 1. For example, soil has different sized aggregates at the surface, stones occur in the field, plant residue may be fixed to, or lying on, the soil and ripples may occur intermittently during sediment transport (and that is to say nothing of intermittent

crusts which may change roughness). The magnitude of R<1 (which may change over time, between events due to change in the roughness 'canopy') is the omitted uncertainty in the authors' methodology.

For clarity, I think the authors should state clearly that they are assuming u*=us*. I think the authors must then ideally estimate, or at least approximate, the uncertainty of making that assumption (R=1) under field conditions when R<1. That 'model' uncertainty will then manifest as an error on u*. When the PSDs are aggregated under this explicit approach, that 'model' uncertainty (not to be confused with the standard deviation of u* already included by the authors) will demonstrate the extent to which there is a difference in the PSDs which is detectable. Any difference between the PSDs remaining after that 'model' uncertainty has been included will have accounted for the dependency of PSD on us* and R.

I think the same issue of uncertainty occurs with the relation between the dependency of emission-dust PSD on u* and the boundary-layer stability. Where u* is based on field measurements, I think it requires the same (as above) expression of uncertainty. As above, this uncertainty is required due to the assumption that u*=us* when field conditions introduce uncertainty. Consequently, the results in the second half of the paper need to be similarly qualified with this uncertainty.

The issue is brought to sharp focus by considering Eq. 8 of the manuscript. The sediment transport Q is related incorrectly to u*^3 (Webb et al., 2020). As above, the available energy for transport is us*^3. Whilst there are conditions when u*=us* and therefore R=1, in the field it is very unlikely that R=1. In this case, the uncertainty of the 'model' assumption u*=us*, needs to be considered (R<1). With this additional uncertainty the arising figures and interpretations may need to be re-evaluated.

Manuscript structure

I think the manuscript mixes unnecessarily theory with results. I think the theory (Eqs. 4, 5, 6 etc. and related text) should be moved to the Methods section. In that Methods

section I think it would be worthwhile explaining carefully how the parameter values of the modelling were chosen (rationale and assumptions) so that it is clear to the reader how the results have been produced.

I find it strange not to have a Discussion section. I wonder if much of the detail in the Introduction would be better moved to the Discussion and then extended as necessary with additional context for the discussion. This would also help the Introduction quickly move the reader through the main issue.

Reference

Webb et al. (2020). A note on the use of drag partition in aeolian transport models. Aeolian Research, 42: 100560.

---

## Author Comment (AC1) · 25 Jun 2020

General comments: The size distribution (emission PSD) of the dust particles emitted from arid and semi-arid surfaces is crucial for the assessment of their numerous environmental impacts. However, this question has been much debated in the last two decades. Some experimental and theoretical studies emphasized the importance of the kinetic energy (increasing with the speed of the wind) of the saltating sand grains for this PSD, whereas others maintained that the PSD was constant. In addition, a quite recent study also showed that the stability of the atmosphere played a role: the more unstable the lower atmosphere, the richer the PSD in fine particles. The aim

of the present study is to re-analyze the data of the JADE experiment carried out in Australia for answering the following questions: 1) does the emission PSD depend on wind speed or not? 2) is it confirmed that the atmospheric stability plays a part?, and 3) why? For answering these questions, the authors select two contrasted events of the JADE field campaign: one that occurred during a hot day (Event 10) with unstable conditions, and another (Event 11) that happened immediately after but in stable nocturnal conditions. Then, they analyze separately the influence of 1) u* (and its probability distribution), and w* (a proxy of atmospheric stability) on the emission PSD. My opinion is that the demonstration of the authors is quite convincing, well organized, and their methods are scientifically sound. Finally, the study brings final answers to ancient questions: yes, the emission PSD does depend on wind speed (through the kinetic energy of the saltators), and yes atmospheric instability promotes the ejection of finer particles (because of the widening og the u* distribution as turbulence increases). In summary, this paper is a very important one and undoubtedly deserve publication in ACP after the few following comments have been addressed:

Response: We are most grateful to Referee 1 for the encouraging comments and constructive suggestions. Referee pointed out the usefulness of the study in clarifying the issue of particle size dependency on wind shear stress and possibly atmospheric boundary layer stability. There are several suggestions we will consider and modify the text accordingly.

Comments and minor concerns:

1) P. 1, line 26: F increases with Q, but it is not simply proportional to it. For a given surface, the so-called 'sandblasting-efficiency' (ratio of F to Q) is usually found to vary with u*.

Response: We will check this. Existing data do suggest that Q~u* cubed, but F~u* to power n and n is not necessary 3. This issue has been clarified in several earlier studies by the first author, together with Dr. Hua Lu, but we will modify the text to be

more precise.

2) P. 2, lines 45 and 48; correct the years for Kok's publications.

Response: Sorry about this. We will check and correct.

3) P. 2, line 62: in Ishizuka et al. (2008) and in this paper, u* is calculated over periods of 1'. This is too short to integrate the time-scales of turbulence in the surface layer (Dupont el al., 2019), but presents the advantage of following more closely the variations of the instantaneous wind speed to which saltation responds in quasi real-time. Conversely, Khalfallah et al. (2020) calculated u* over longer periods of 16'. The smoothening of the u* statistical distribution resulting from this averaging probably explains that they could not detect any notable influence of u* on the emission PSD.

Response: We would agree with the referee. For boundary-layer studies, to establish a flux-gradient relationship (here shear stress and mean wind), averaging over 1 min would not be enough. But for saltation, 1 min average gives the advantage to examine the fluctuations of saltation and dust emission. It is indeed important to have another look at this problem and see whether the comparison of this study and that of Khalfallah et al (2020) is a fair comparison.

4) P.3 line 64: JADE (not JADA)

Response: Thanks, we will correct.

5) P.4, line 88: In sedimentology, the texture of a soil is defined from the size-distribution of its particles after full dispersion. So, the soil is sandy loam.

Response: This is a good point. It seems that we need to have a look in the study of sedimentology. We have misunderstood this (may be for years).

6) P.6: Figure 4a. In the insert, the same color code as in the rest of the figure could be used to differentiate Event 10 from Event 11.

Response: Thanks. We will check this.

7) P.6; line144: 'do not substantially differ'. What does this mean?

Response: Indeed, this is a bit sloppy. We will try to be more precise.

8) P.8; line 177: what are the implications of the fact that the distribution of $\tau$ is skewed to smaller values?

Response: We will add a line here to be more specific. It seems to suggest that the LES results of Klose et al. (2014) seem to be qualitative reasonable.

9) P. 10, line 208: experiments 'were'

Response: Thanks. We will correct it.

10) P. 113, lines 229-235: Please rewrite this part to avoid repetitions and clumsy formulations.

Response: Thanks. We will try to improve.

---

## Author Comment (AC2) · 25 Jun 2020

This paper addresses two key-questions that are debated by the scientific community working on wind erosion and atmospheric dust. The first question, debated from several years, concerns the dependence or not of the emission-dust PSD with the wind friction velocity. The second one, more recently laid on the table by Khalfallah et al. (2020), deals with the dependence of the same emission-dust PSD with the atmospheric boundary-layer stability. These questions are under debate mainly because obtaining relevant observations in natural conditions to investigate such dependencies is difficult. Long and complex campaigns are required for this, a drastic selection among

the observations is necessary to isolate the best situations and, therefore, relevant data are scarce. Thus, one of the first interest of this paper is to propose a re-analyze of the data from the JADE campaign (Ishizuka et al., 2008). Beyond that, this article offers an original and in-depth analysis of these data. Finally, it proposes numerical simulations to support the conclusions. More precisely, the paper provide convincing observations on the dependency of emission-dust PSD with the wind friction velocity and atmospheric boundary-layer stability. The experimental results reported in Figures 4, 5 and 6 are the key figures of this paper. The discussion on the PDF of u* and on its role on the intensity of the saltation is really convincing. The authors also propose interesting and elegant explanations for these dependencies by examining the role played by the wind friction velocity PDFs in cases of high and low u* and of different stability conditions. I am not sure that this paper will definitively close the debate (more experimental data will probably be necessary for this) but I am sure that this document is an important contribution in order to better understand what controls PSD dust emissions and to elaborate future experiences. The paper is well written and structured even if additional information is needed in some places. I recommend to publish this paper with minor corrections.

Response: We are most grateful to Referee 2 for his/her review and helpful comments, which we will address in revision. First of all, we fully agree with the referee, that this paper is part of the ongoing debate and we definitely need more experimental data to fully solve the question. This paper points to the PSD dependency on u* and atmospheric boundary layer stability. In fact, the Japanese team (Ishizuka et al.) has more recently collected data in Mongolia. We will need more time to process the data.

1. Introduction: The introduction is very well written, concise and gives a clear idea of the scientific context. lines 41-42 (and also mentioned on line 144-145): I agree with the authors that dust-airborne PSD measured very close to the surface can probably be assimilated to dust-emission PSD by assuming that the difference of the particle diffusivity compared at that of other scalar (and thus, its dependency on particle diameter) can be neglected. However, because this assumption is an important point of the paper, it deserves to be better discussed, especially because there are few experimental data on particle diffusivity, especially for small particles, and that most of the information we have come from models. In the same way, since the authors indicate that size-resolved dust fluxes were measured during JADE (line 67) a comparison between dust–airborne PSD and dust-flux PSD should be added to support this assumption, at least in the supplement. line 49: the only reference to the Pisso et al., 2019 's paper is not sufficient to support the statement that "The proposed emission-dust PSD is frequently used in dust models". Line 52: replace airborne-dust PSD by "dust flux PSD" or "emission dust PSD"

Response: This is substantial suggestion. We will have to look into this again. The problem is that "dust PSD at emission" is never measured to our best understanding. One way out of this may be to show the difference of the PSD on the different levels, which may indirectly support the claim that diffusion cannot the main reason for dependency of PSD on u* and ABL stability.

2. JADE data line 64: replace JADA by JADE. I know that the JADE experiment has been described in details in various papers, especially in Ishizuka et al., 2008, and does not need to be described again. However, the presentation of the data in this paper is too short: for example, it is never indicated that OPCs measured number concentrations that are further converted into mass concentrations assuming spherical particles of density= 2380 kg m-3. In the same way, Ishizuka et al. (2014) mentioned that the measurements for the bin 0.3-0.6 _m were not considered because of a significant difference between the two OPCs for this bin. In the present paper, this bin is used and the reason for that should be explained, at least to be consistent with the previous paper. However, it must be noted that this point is not critical for this paper since the contribution of this bin is negligible in mass as shown on figures 3 and 4.

Response: Thanks for this insight. Indeed, there are some issues related to the accuracy of the 0.3-0.6 um size bin. This issue is more important if we use the flux for

this bin to validate the dust emission model. As for the airborne PSD examination, the inaccuracy does not seem to be some important, because, as the referee correctly pointed out, its contribution to the PSD is small. We will probably modify the text and the graph, to show that there is an issue here.

In the same way, the data from OPC measuring at 2 m high were not considered as relevant in Ishizuka et al. (2014) (because they does not correlate with the OPC measurements performed at the other heights) but are used in this paper. This should also be explained.

Response: We had another look at the PSD measured at 2m, they seem to be consistent with the PSDs measured at 1 and 3.5m. The first author decided to include it (against the suggestion of Dr. Ishizuka to exclude this). But of course, we can take the 2m PSD out, and the basic results will not change, apart from very fine details.

Finally, line 96, it is indicated that no rainfall occurred as a consequence of the cold front crossing but in Ishizuka et al; (2008), it is mentioned that the rain sensor detected several very small precipitation events that have been also observed by the authors. Even if, as mentioned in Ishizuka et al. (2008) the drying of the soil was very rapid, this should mentioned.

Response: The authors will re-discuss on this point to clarify.

Figure 1: The measured dust concentrations are very high (several mg m-3!) even for event 11. This should be underlined in the text. Indeed, since the authors used airborne-dust PSD and not dust flux PSD, it is necessary to provide arguments showing that the measured dust PSD can be directly linked to dust emission. And such high concentrations of dust strongly suggests that the contribution of advection to the measured dust PSD is probably very limited.

Response: Very close to the surface during dust event, spikes of very high dust concentration seem to be possible. Plotted are the data as collected. But we will recheck

this to be sure.

line 88: reformulate. A soil has only one texture. A formulation such as "The results of the analyze by method A correspond to (or suggest) a loamy sand texture while the results of the analyze by method B..." should be better.

Response: This comment is consistent with Referee 1's comment. We will change accordingly.

line 91: I appreciate the method consisting to select among various dust events those the best adapted to the objective of the study. However, the justification for selecting these two events is very short ("Event-10 occurred under daytime unstable, while Event-11 under night-time stable, conditions"). I imagine that they were other events among the 12 aeolian events recorded during JADE that occurred also in stable or unstable conditions. Are these events selected because the stability conditions were particularly constant for these two events?

Response: Yes. We fully agree. Events 10 and 11 are bested studied and are most strikingly different. The data for the other events are less complete and it is a really a matter of labor to process all the data in great detail. But we will look into this.

3. Results Figure 3: we have no idea of the number of points contributing to each u* category. This should be added in the legend of the figure; in the same way, no information is provided on the spread of the different points that allow to construct the PSD (standard deviation bars should be added). The authors write "that dust PSDs for Event-10 and -11 considerably differ": maybe an additional panel reporting the difference between PSD10 and PSD11 for similar u* categories could better illustrate these differences in PSD.

Response: Thanks for this suggestion. We need to look into the data to provide the necessary information the referee has requested.

Figure 4: This figure in which are averaged all heights and all u* is important since it

clearly shows that the dust and saltation PSD shift between the two events. However, this averaging approach is not well introduced and it should not be obvious for the reader to understand why it is useful and relevant to make such averaging. The paper should explain that. The insert is too small and should be a figure by itself. Same comment concerning standard deviations as for figure 3.

Response: Again, thanks for this suggestion, and we will do the explanations in the revision.

On figure 6, there is a shouldering in the high values for the observed u* distribution corresponding to event 11. This suggests that the u* PDF could be bimodal for this event. Moreover, the adjusted Gaussian PDF for u* does not include this shouldering reducing the variance of the u* Gaussian PDF for event 11. This should be discussed. The numerical simulations are interesting and illustrate the sensitivity of the impact kinetic energy to different parameters on which the stability conditions could act. They clearly suggest that larger variances in u* PDF, as generally observed in unstable conditions, generate stronger saltation and thus should be responsible for higher production of fine particles.

Response: In the very earlier phase of Event 11, there are some big u* values which produced the "shouldering". It is probably not that Event 11 is generally a case with bimodal u* values. We agree, we need to look into this and see what implications the steadiness of the process has on the results.

---

## Author Comment (AC3) · 26 Jun 2020

**Overall assessment**

I read the manuscript from the perspective of a specialist in wind erosion and dust emission but a generalist to this specific focus of the manuscript. The authors provide a clear, logical development of the focuses of their manuscript on the different bases for explaining particle size distributions of emitted dust and the dependency of air borne dust PSD on atmospheric boundary layer (ABL) stability. The topic is valuable for the community and the work is well presented. However, I am not convinced by the approach used in the manuscript. I think the work in the manuscript omits uncertainty. If

that uncertainty were included, I think it may lead to different / alternative conclusions. Therefore, to increase confidence in the results I think the omitted uncertainty must be tackled, in some form or other, before the work can be published. I provide below additional information on this point. I also think that some improvements in the structure of the manuscript will help the reader more easily follow the explanation of the work. In short, I am thoroughly supportive of the work. I think the manuscript needs to be revised to give confidence that the results are indeed detectable and therefore interpretations are robust to the uncertainty. The nature of the revisions I describe below I think, are consistent with a major revision, despite not being too difficult to achieve in a short period of time, if all other things were equal.

Response: Many thanks for Referee 3 for the constructive comments. She/he emphasized on the uncertainty of the analysis, and uncertainty related to roughness correction for saltation. Uncertainty analysis is always important, but the PSD difference we detected is mainly based on field observations and the differences are systematic. Investigation on the significance of the difference is possible, however, we pretty sure that the difference is statistically significant.

Main issues

Wind friction velocity uncertainty In the abstract, it is stated that friction velocity $u^*$ is a surrogate for surface shear stress and descriptor for saltation bombardment intensity. Line 32 states that "for a given soil, the particle size distribution of dust at emission (emission-dust PSD), $p_s(d)$, must depend on saltation bombardment or on friction velocity". The JADE field measured data are used to show that the (finely resolved) particle size distribution is dependent on measured wind friction velocity. In contrast to this approach, it is well known (cf. sediment transport models) that the wind energy available for saltation bombardment is not $u^*$, it is the energy $u_s^*$, which remains after wind momentum has been extracted by the roughness 'canopy'. In other words, $u_s^*=u^*.R$ where R is the partition of drag. Under controlled conditions with homogeneous material, smooth surface (bed) without ripples and the bed reset after each experiment,

it may be reasonable to assume that R=1. However, the authors use field data which, even under the smoothest field conditions are very likely to cause R not equal to 1. For example, soil has different sized aggregates at the surface, stones occur in the field, plant residue may be fixed to, or lying on, the soil and ripples may occur intermittently during sediment transport (and that is to say nothing of intermittent crusts which may change roughness). The magnitude of R<1 (which may change over time, between events due to change in the roughness 'canopy') is the omitted uncertainty in the authors' methodology. For clarity, I think the authors should state clearly that they are assuming u*=us*. I think the authors must then ideally estimate, or at least approximate, the uncertainty of making that assumption (R=1) under field conditions when R<1. That 'model' uncertainty will then manifest as an error on u*. When the PSDs are aggregated under this explicit approach, that 'model' uncertainty (not to be confused with the standard deviation of u* already included by the authors) will demonstrate the extent to which there is a difference in the PSDs which is detectable. Any difference between the PSDs remaining after that 'model' uncertainty has been included will have accounted for the dependency of PSD on us* and R. I think the same issue of uncertainty occurs with the relation between the dependency of emission-dust PSD on u* and the boundary-layer stability. Where u* is based on field measurements, I think it requires the same (as above) expression of uncertainty. As above, this uncertainty is required due to the assumption that u*=us* when field conditions introduce uncertainty. Consequently, the results in the second half of the paper need to be similarly qualified with this uncertainty. The issue is brought to sharp focus by considering Eq. 8 of the manuscript. The sediment transport Q is related incorrectly to u*ËĘ3 (Webb et al., 2020). As above, the available energy for transport is us*ËĘ3. Whilst there are conditions when u*=us* and therefore R=1, in the field it is very unlikely that R=1. In this case, the uncertainty of the 'model' assumption u*=us*, needs to be considered (R<1). With this additional uncertainty the arising figures and interpretations may need to be re-evaluated.

Response: About the issue related to roughness correction, we are not sure whether

it is relevant here, because using field measurements to detect changes in surface roughness (e.g. ripples) will be difficult, and will probably lead to more uncertainties in saltation estimates. But as saltation flux used in this study is measured, we do not see how the roughness correction comes into play. What can be done is probably some sensitivity tests on how erosion modified aerodynamic roughness length causes different saltation intensity, but then this is already done in the recent paper by Webb et al.

Manuscript structure

I think the manuscript mixes unnecessarily theory with results. I think the theory (Eqs. 4, 5, 6 etc. and related text) should be moved to the Methods section. In that Methods section I think it would be worthwhile explaining carefully how the parameter values of the modelling were chosen (rationale and assumptions) so that it is clear to the reader how the results have been produced. I find it strange not to have a Discussion section. I wonder if much of the detail in the Introduction would be better moved to the Discussion and then extended as necessary with additional context for the discussion. This would also help the Introduction quickly move the reader through the main issue.

Response: Thanks for the suggestions on structure. We will try to improve the manuscript accordingly.
* * *

---

## Author Comment (AC4) · 26 Jun 2020

We are most grateful to Dr. Sylvain Dupont for carefully reading our manuscript and providing insightful and detailed comments for discussion. These comments are very valuable for us to improve our paper and approaching the truth.

Dr. Dupont first commented on the results of the Khalfallah et al. (2020) paper and pointed out possible discrepancies between his own analysis and the results presented in Khalfallah et al. Indeed, this paper is triggered by the study of Khafallah et al. which helped us to decide to finally have a thorough look at the PSD issue. We thought this issue was resolved until the questions raised by the Kok's paper which has caused a

stir in the dust research community. Now, our results show the PSD at dust emission is u* dependent. This seems also to be the view of Dr. Dupont. With respect to PSD dependency on atmospheric boundary-layer stability, our results seem to support the finding of Khalfallah et al. qualitatively, but we have some considerations of their interpretation why this might be so. We are not convinced that "diffusion" caused this dependency.

As we do not know exactly, how colleague Khalfallah et al. processed their data, we cannot judge the reliability of their conclusions, but Dr. Dupont is in a much better position to make the judgement, as he works with the authors of the afore-mentioned paper. With the insight Dr. Dupont provided, we will modify in the revised paper and to be more cautious with the interpretation of the results of Khalfallah et al., although we have certainly tried not to "rely" on their work.

The second point of Dr. Dupont is important, namely, that he is convinced of PSD dependency on u*, not necessary on ABL stability. Our view is somewhat different. In our paper, we have tried to make it clear that there is a mean u* and a u* variance, the PSD of dust at emission is not only dependent on the u* mean but also on the u* variance. This dependency arises because the saltation bombardment is non-linearly dependent on u*. In essence, this is the problem of saltation/dust emission intermittency. In a series of related studies (e.g. Klose et al. 2014), we have been considering how turbulence causes dust emission. Suppose the mean u* equals to u*t, then there would be no saltation and no saltation bombardment, but if u* has a distribution, then intermittent saltation and saltation bombardment will occur, and the PSD of dust at emission will be dependent on the PDF of u*. This really is the main point of this study, and the idea is already in several Klose et al. papers.

Now, does turbulence intensity (actually the PDF of u*) depends on ABL stability? We think so, as the large-eddy simulation of Klose et al. (2014) shows and also the JADE observations. We have carried out recently a wind-tunnel experiment, again showing the dependency of u* PDF on ABL stability and the strong impact on dust

emission. The results of the wind tunnel experiments will be summarized and send for consideration of publication.

Thanks for mentioning the Kaimal and Finnigan (1994) book (Yaping Shao and John Finnigan have worked in the same group for some years and is one of the first readers of the book). The r_uw curve in Fig 1.9 of KJ (1994) book does not seem to apply here, because (1) there is nothing said about the variance of the shear stress only the mean; (2) it only states that shear stress normalized with the wind variances is fairly constant (not exactly constant, actually why not exactly?); (3) earlier measurement of shear stress was mainly down somewhere in the ABL at some level, not really at the surface; and (4) their Fig 1.10 actually shows that the variance of wind is dependent on ABL stability (i.e. the variance normalized with w* is fairly universal).

We agree with Dr. Dupont, we probably need to do more cases, as Ref. 2 also mentioned, but the other JADE cases are less complete and much more work is needed to process the data.

Dr. Dupont made several very nice suggestions.

(1) More figures for characterizing the events: to understand what happened during the erosion events 10 and 11, show time variation of dust and saltation PSD during the events. This is a good idea. We will look into this. (2) PSD of emitted dust flux: This seems to be difficult. Such a PSD was not observed. (3) Condensation: as far as we know, there was no condensation, but there were a few drops of rainfall accompanied with the cool change, although no rainfall was recorded. We will have to look into this to be able to answer. (4) Enhanced cohesion in night: this is an interesting point, but it is difficult to validate. In a separate study by the first author (unpublished), he is working on the modelling of soil moisture under extremely dry conditions. (5) Surface modified by saltation: Yes. This is likely, but we cannot validate this. As Dr. Dupont suggests, we can have a look at the saltation PSD evolution. (6) First justifications: Dr. Dupont is right. It should be interpreted differently. (7) u* variance: This is an important

point. As Dupont et al. (2018) shows that u* needs to be averaged over 15 to 30 mins for the flux-gradient relationship. However, there is no doubt that shear stress fluctuates due to large eddies, and shear stress has a PDF. This PDF is important to dust emission which rapidly responses to surface shear stress. The selection of 1 min for shear stress averaging seems to be reasonable, this is to assume that saltation can reasonably respond to shear stress variations on this scale. This is the whole point of this paper. We are willing to debate with Dr. Dupont on this in greater detail. (8) Saltation bombardment intensity: We have discussed this above. KF (1994) book, Fig. 1.9 states r_uw is almost independent of z/L, but r_uw is shear stress normalized with flow velocity variance which do vary strongly with stability, as their Fig. 1.10 shows. But we would agree with Dr. Dupont, this is an unsolved issue, because we do not fully understand how the laminar layer close to the surface behaves. There are theories about the possibility that the laminar layer breaks up. Our unpublished wind-tunnel experiment (measuring shear stress using Irwen sensors showing the fluctuations of shear stress related to large eddies). Again, the whole point is that we have to move away from the tradition "mean" flow characteristic approach and consider more the PDF of the turbulence quantities, which are important to understanding the PSD of dust at emission.

There seems to be a misunderstanding somewhere. What we try to say in justification 3 is actually that the diffusion aspect due to enhanced or not enhanced turbulence with respect to instability does not affect the saltation trajectory too much. In this sense, Dr. Dupont is right. But the initial velocities of the saltation particles seem to be important.

It is great to discuss with colleague Dr. Dupont.

---

## Short Comment (SC2) · 10 Jul 2020

**Dependence of particle size distribution at emission on friction velocity**

I thank the esteemed authors for writing this stimulating manuscript on a topic that has been of some controversy within the dust community: does the size distribution of emitted dust depend on wind properties and, if so, how? This manuscript aims to address two more specific sub-questions, namely (1) does the emitted dust size distribution depend on wind friction velocity and (2) does the emitted dust size distribution depend on atmospheric boundary layer stability? My colleague Sylvain Dupont submitted an excellent comment on this second part of the manuscript; my comment focuses on the first question.

I value the contribution of this manuscript and hope that it can be published after revisions. The authors rightly identified that the question of whether emitted dust PSD depends on wind speed (and stability) should be investigated further as there is some contradictory evidence. After reading the paper carefully, I think there are a few issues in the present work that I would recommend addressing.

- As Sylvain Dupont also pointed out, a key methodological issue is that the authors equate airborne PSDs with the emitted dust PSD. This might be problematic because the size distribution of airborne aerosols is a sum of dust emitted at the measurement site, dust emitted upwind from the measurement site, and aerosols of any species advected to the measurement site. Additionally, the PSD of dust advected from upwind is known to depend on u\* (see for instance Dupont et al., (2015)), so using the airborne PSD could create a change in the airborne PSD even if the emitted dust PSD remains constant. To avoid this problem that otherwise could shed doubt on the results, I recommend the authors either use the emitted dust flux inferred from the gradient method (Gillette, Blifford, and Fenster 1972) or should provide strong support in this paper that "In JADE, airborne-dust PSD [...] well represent the emission-dust PSD" (line 57-8) and that this very strong match does not change with u\*.
- Second, this work reanalyzes JADE data that was previously analyzed in an excellent paper by Shao et al. (2011). The present paper reaches rather different conclusions on the question of whether the emitted dust PSD depends on wind speed, yet does not note this seeming discrepancy anywhere that I could find. Indeed, Shao et al. (2011) report no dependence of the dust PSD on u\* (their Fig. 12, copied below) and note that (their p. 15): "A clear and systematic dependency of psd on u\* cannot be identified". I thus recommend that the revised paper (1) articulates the reasons why a reanalysis of the JADE data is necessary since these data were already analyzed in Shao et al. (2011), and (2) clearly explains why the present results are different from those reported in Shao et al. (2011) and why the authors think the analysis here is correct and, seemingly by implication, why the analysis and conclusion in Shao et al. (2011) is incorrect.
- Related to this previous point, I recommend that the authors use a broader range of data, similar to Shao et al. (2011). Their Figure 3 is similar to Figure 12 in Shao et al. (2011) in that it shows the dust PSD for different u\* values, but it is subsetted to include only two events (10 and 11) whereas Figure 12 in Shao et al. (2011) presumably includes data from all measured events.

This might raise concerns that the conclusions of this study would change if more data than just the two events were considered.

• I think it would be beneficial to represent previous work better. As the authors are no doubt aware, the question of whether the emitted dust size distribution depends on wind friction velocity has been the subject of quite a number of studies, and several of those studies are not cited here. These include Fratini et al., 2007, Shao et al. 2011, and Huang et al., 2019. I'd also recommend noting more explicitly that the majority published measurements do not find a dependence of emitted dust size distribution on wind speed (see figure below and analyses in Kok (2011b) and Mahowald et al. (2014)). I think that context is important for the reader to interpret the opposite finding in this paper.

---

## Author Comment (AC5) · 12 Jul 2020

We greatly appreciate Colleague Dr. J. Kok for his comments. That Dr. Kok took time to provide such thoughtful comments shows the need to clarify the dust PSD issue. We have not discussed in great details yet, but will simply provide a quick reply.

First, "Airborne PSD as emitted dust PSD": to our best knowledge, dust emission PSD has been directly observed. All dust emission PSDs reported are airborne dust PSDs. We welcome our colleagues to correct us, if we are wrong. The JADE airborne dust PSDs are of good quality and are probably close(r) to dust emission PSD.

[Figure]

The argument that dust advection depends on u* is interesting, but does not seem to apply here. Advection is $\sim u \partial C/\partial x \sim u\_* \, \partial C/\partial x$. In case of weak dust concentration gradient, advection does not play a major role. The JADE site is fairly homogeneous and the dust PSDs are measured close to the surface. Therefore, we can safely exclude the influence of advection on dust PSD.

Second, "Consistency of Evidence": Fig. 12 of Shao et al. (2011) was included at recommendation of a referee. It may be that, in that analysis, too much averaging and too small u* intervals blurred the dependency of dust PDS on u* and ABL stability. We did not carefully examine the individual runs as we are doing now. Colleague Dr. Kok and others have made an excellent suggestion, we do need to have a look at the statistical significance of the results.

Statistical significance test is generally lacking in dust related studies and this is partially why we have so much confusion in aeolian research.

Third, earlier results: I like this suggestion of Dr. Kok very much. But, to be honest, this is difficult, as it is hard to get to the bottom of the various data sets. I believe Dr. Kok and colleagues have properly estimated the error margins of the previously published data and the averages may pointing to "universal dust PSD". However, I still think, to better understand the physics, we need to do case studies.

Fourth, statistics: This is a very good suggestion.

5th Line 30-32: "Since inter-particle cohesion depends on particle size, d, the fraction of dust emitted must also depend on d. Thus, for a given soil, the particle size distribution of dust at emission (emission-dust PSD), ps(d), must depend on saltation bombardment or on friction velocity" and line 140-1 "u* is a descriptor of saltation bombardment intensity". This argument implicitly assumes that the impact speed of saltating particles increases with the friction velocity. It is highly intuitive that it would, but there is a very solid body of research that indicates that particle impact speed actually does not depend on friction velocity for transport-limited saltation. This lack of

dependence of particle speed on wind speed was first proposed by Ungar and Haff (1987) because particle-wind feedbacks force an approximately constant saltator impact speed. It has since been confirmed by a large body of experimental (e.g., Namikas (2003), Rasmussen and Sorensen (2008), Creysells et al. (2009), Ho et al., (2011), Martin and Kok (2017)) and numerical (e.g., Duran et al. (2011), Kok et al. (2012)) work. The authors can of course present evidence to support their viewpoint counter to this literature, but I recommend acknowledging this extensive literature.

This is interesting. Let us make two thinking experiments. Exp 1: u* = u*t, particle creeps and has impact velocity 0. Exp 2: u* > u*t, particle saltates and has impact velocity larger than 0. This shows particle impact depends on u*. But thanks for pointing out the study which conclude differently. We will have to learn how it is possible that impact is u* independent.

6th Line 48-9: "Kok (2011a, 2011b) then proposed an emission-dust PSD and estimated its parameters from airborne-dust PSDs." That's actually not quite correct: Kok (2011a) only used emitted dust size distribution because airborne-dust PSDs are a convoluted sum of emission and advection (see comment above and by Sylvain Dupont). Also, the years on the references are incorrect (I corrected them in the quote above).

Thanks. We will check this.

7th I'm a bit confused how to interpret the 0-0.25 m/s u* category in the present paper's Figure 3, as this would include events without saltation where dust is not actively emitted but only advected. I suspect the authors are only using data for which saltation was occurring. If so, I recommend that the authors note that. And if not, I recommend the authors subset the data to only include active saltation data.

Saltation is intermittent and occurs below 0.25 m/s u*. This is a point we try to make, namely, turbulence (and saltation intermittency) plays an important role in dust PSD. It seems that this point did not come cross clearly, as this also appears to be the

impression of colleague Dr. Dupont.

Many thanks to Colleague Dr. J. Kok.

---

## Author Response (AR1)

**Point-by-point response**

**Reply to Review 1:**

We are most grateful to Referee 1 for the encouraging comments and constructive suggestions. The Referee pointed out the usefulness of the study in clarifying the issues of particle size dependency on shear stress and possibly atmospheric boundary layer stability. We have now considered the suggestions by the Referee and modified the text accordingly.

*C1) P1, l26: F increases with Q, but it is not simply proportional to it. For a given surface, the so-called 'sandblasting-efficiency' (ratio of F to Q) is usually found to vary with $u_*$.*

Existing data suggest that $Q \sim u_*^3$, but $F \sim u_*^n$ and n is not necessary 4. This issue has been clarified in several earlier studies by the first author together with Hua Lu. Different definitions of "saltation bombardment efficiency" exist in the literature, but the Referee is certainly right in stating that F/Q varies with u*. Indeed, this has been one of the main conclusions of Shao (2001; 2004). In order not to slow down the pace of introduction, we added a footnote for clarification.

*C2) P2, l45 and 48: correct the years for Kok's publications*

Corrected.

*C3) P2, l62: in Ishizuka et al. (2008) and in this paper, u* is calculated over periods of 1'. This is too short to integrate the time-scales of turbulence in the surface layer (Dupont el al., 2019), but presents the advantage of following more closely the variations of the instantaneous wind speed to which saltation responds in quasi real-time. Conversely, Khalfallah et al. (2020) calculated u* over longer periods of 16'. The smoothing of the u* statistical distribution resulting from this averaging probably explains that they could not detect any notable influence of u* on the emission PSD.*

We would agree with the referee. In boundary-layer studies, using the flux-gradient relationship to estimate friction velocity from wind profile requires relatively long time averaging, typically 15 to 30 minutes, and averaging over 1 min would not be enough. But for saltation, 1-minute average gives the advantage to examine the fluctuations of saltation and dust emission. To include turbulence in the study of dust emission particle size is an important aspect of our paper. We do not know whether Khalfallah et al. (2020) would reach a different conclusion if $u_*$ is averaged shorter. Dr. Alfaro informed us that they are reexamining their data. Discussions added to the text.

*C4) P3 l64: JADE (not JADA)*

Thanks. Corrected.

*C5) P4 l88: In sedimentology, the texture of a soil is defined from the size-distribution of its particles after full dispersion. So, the soil is sandy loam.*

Thanks for both Referees pointing out this. The first author has misunderstood this for years. It is now corrected.

*C6) P6, Fig. 4a: In the insert, the same color code as in the rest of the figure could be used to differentiate Event 10 from Event 11.*

Changes made as suggested. Thanks.

*C7) P6, l144: 'do not substantially differ'. What does this mean?*

This line is deleted at this location, but we have added a substantial section to discuss the problem of data uncertainty.

*C8) P8, l177: what are the implications of the fact that the distribution of ï¸At' is skewed to smaller values?*

This suggests the LES results of Klose et al. (2014) seem to be qualitative reasonable.

*C9) P10, l208: experiments 'were'. . .*

Changed. Thanks.

*C10) P113, l229-235: Please rewrite this part to avoid repetitions and clumsy formulations.*

We tried to improve, but are not sure how to write much better.

**Reply to Review 2:**

We are most grateful to Referee 2 for his/her review and helpful comments, which we have addressed in the revision.

First of all, we fully agree with the referee, that this paper is part of the ongoing debate and we definitely need more experimental data to fully solve the question. This paper points to the PSD dependency on u∗ and atmospheric boundary layer stability. In fact, the Japanese team (Ishizuka et al.) have more recently collected data in Mongolia. We will look into that dataset, once it is ready. But more time is needed to quantity check, preprocess and homogenize the data.

*C1): Introduction: The introduction is very well written, concise and gives a clear idea of the scientific context. lines 41-42 (and also mentioned on line 144-145): I agree with the authors that dust-airborne PSD measured very close to the surface can probably be assimilated to dust-emission PSD by assuming that the difference of the particle diffusivity compared at that of other scalar (and thus, its dependency on particle diameter) can be neglected. However, because this assumption is an important point of the paper, it deserves to be better discussed, especially because there are few experimental data on particle diffusivity, especially for small particles, and that most of the information we have come from models. In the same way, since the authors indicate that size-resolved dust fluxes were measured during JADE (line 67) a comparison between dust–airborne PSD and dust-flux PSD should be added to support this assumption, at least in the supplement. line 49: the only reference to the Pisso et al., 2019 's paper is not sufficient to support the statement that "The proposed emission-dust PSD is frequently used in dust models". l52: replace airborne-dust PSD by "dust flux PSD" or "emission dust PSD"*

These are substantial suggestions. The Referee pointed out the need for discussion of heavy particle diffusivity in turbulent flows. This is a large topic in itself and studies dedicated to the topic are numerous. The Referee is right in saying that "there are few experimental data on particle diffusivity, especially for small particles, and that most of the information we have come from models". To our best knowledge, the understanding of particle diffusivity still rests on the study of Csanady (1963, Turbulent Diffusion of Heavy Particles in the Atmosphere. J. Atmos. Sci. 20, 201–208). His result is that particle eddy diffusivity $K_p$ is related to eddy diffusivity $K$ by

$$K_p = K\left(1 + \beta^{2} w_t^2/\sigma^2\right)^{-1/2}$$

where $\beta$ is a coefficient (about 1 to 2), $w_t$ is particle terminal velocity and $\sigma$ is the standard deviation of turbulent velocity. The studies of Csanady (1963), Walklate (1987, A random-walk model for dispersion of heavy particles in turbulent air flow. Boundary-Layer Meteorol. 39:175–190) and Wang and Stock (1993, Dispersion of heavy particles by turbulent motion. J Atmos Sci 50:1897–1913) all suggest that $K_p$ depends on particle size and turbulence intensity. For the particle-size range we consider, $K_p/K$ is close to 1 for $\sigma = 0.5$ m/s, and for $\sigma = 0.1$ m/s, it is still larger than 0.95 (Shao, 2008, Physics and Modelling of Wind Erosion, Fig. 8.12).

However, the variation of airborne-dust PSD in the vertical direction due to transport processes in the atmosphere cannot be ruled out. Plotted here is a comparison of (Event-10 averaged) airborne-dust PSD measured at 1, 2 and 3.5m. Ishizuka et al. (2014) excluded the 2m OPC data, because they do not correlate with the OPC measurements at the other heights. We had another look at the airborne-dust PSD measured at 2m, it does show some differences from those measured at 1m and 3.5m. To reduce uncertainties, we will now exclude the 2m OPC data from the study. The figure shows that the airborne-dust PSD observed at 1m and 3.5m also somewhat differ, with peak particle size shifting from smaller to larger values. We cannot fully explain this difference, but suspect that particle deposition may have played a role. The dependency of deposition velocity on particle size in the size range of 1 to 10 μm is rather complex (Zhang and Shao 2014, A new parameterization of particle dry deposition over rough surfaces. Atmos. Chem. Phys 14, 12429-12440).

We are not aware of direct measurements of emission-dust PSD. A flux-dust PSD is possible, namely, a PSD estimated based on size-resolved dust flux. However, because dust flux is estimated from the vertical gradient of dust concentration, the flux-dust PSD describes basically the dependency of concentration gradient on particle size, not the concentration itself on particle size.

[Figure]

We added this graph to the text and added discussions similar to the above to the text. We have cited more studies, in which invariant emission-dust PSD is used.

*2. JADE data line 64: replace JADA by JADE. I know that the JADE experiment has been described in details in various papers, especially in Ishizuka et al., 2008, and does not need to be described again. However, the presentation of the data in this paper is too short: for example, it is never indicated that OPCs measured number concentrations that are further converted into mass concentrations assuming spherical particles of density= 2380 kg m-3. In the same way, Ishizuka et al. (2014) mentioned that the measurements for the bin 0.3-0.6 μm were not considered because of a significant difference between the two OPCs for this bin. In the present paper, this bin is used and the reason for that should be explained, at least to be consistent with the previous paper. However, it must be noted that this point is not critical for this paper since the contribution of this bin is negligible in mass as shown on figures 3 and 4.*

*Finally, line 96, it is indicated that no rainfall occurred as a consequence of the cold front crossing but in Ishizuka et al; (2008), it is mentioned that the rain sensor detected several very small precipitation events that have been also observed by the authors. Even if, as mentioned in Ishizuka et al. (2008) the drying of the soil was very rapid, this should be mentioned.*

We corrected "JADA" to "JADE".

Thanks for the insightful comments. There are some issues related to the OPC 0.3-0.6μm size bin if the data are used for dust flux calculations (Shao et al. 2011). For airborne-dust PSD examination, this does not seem to be a problem. The Referee pointed out, this is not a critical issue. We have made no changes in the text in relation to the 0.3-0.6μm size bin.

As discussed above, the 2m-OPC data are now excluded from the analysis. The overall conclusion of the paper remains qualitatively unchanged.

We modified the text to include the description of Ishizuka et al. (2008) regarding possible precipitation. We added some words in this respect to the discussion section.

*Figure 1: The measured dust concentrations are very high (several mg m-3!) even for event 11. This should be underlined in the text. Indeed, since the authors used airborne-dust PSD and not dust flux PSD, it is necessary to provide arguments showing that the measured dust PSD can be directly linked to dust emission. And such high concentrations of dust strongly suggest that the contribution of advection to the measured dust PSD is probably very limited.*

We have added some text as the Referee suggested and also modified the figure. Advection may have contributed to the extremely high concentrations shortly before and after the cool change. These time periods are now excluded from our analysis.

*line 88: reformulate. A soil has only one texture. A formulation such as "The results of the analyze by method A correspond to (or suggest) a loamy sand texture while the results of the analyze by method B. . ." should be better.*

Referee I made the same comment. We have changed the text as the referees have suggested.

*line 91: I appreciate the method consisting to select among various dust events those the best adapted to the objective of the occurred under daytime unstable, while Event-11 under night-time stable, conditions"). I imagine that they were other events among the 12 aeolian events recorded during JADE that occurred also in stable or unstable conditions. Are these events selected because the stability conditions were particularly constant for these two events?*

We agree with the Referee. In the revised paper, we use Events 10 and 11 for detailed case studies but present all events in the new discussion section.

*3. Results*

*Figure 3: we have no idea of the number of points contributing to each u\* category. This should be added in the legend of the figure; in the same way, no information is provided on the spread of the different points that allow to construct the PSD (standard deviation bars should be added). The authors write "that dust PSDs for Event-10 and -11 considerably differ": maybe an additional panel reporting the difference between PSD10 and PSD11 for similar u\* categories could better illustrate these differences in PSD.*

Thanks for this suggestion. We have now substantially reworked on the data. Presented the averages for the entire JADE dataset.

*Figure 4: This figure in which are averaged all heights and all u\* is important since it clearly shows that the dust and saltation PSD shift between the two events. However, this averaging approach is not well introduced and it should not be obvious for the reader to understand why it is useful and relevant to make such averaging. The paper should explain that. The insert is too small and should be a figure by itself. Same comment concerning standard deviations as for figure 3.*

Thanks for this suggestion. We tried to better explain in the revised version.

*On figure 6, there is a shouldering in the high values for the observed u\* distribution corresponding to event 11. This suggests that the u\* PDF could be bimodal for this event. Moreover, the adjusted Gaussian PDF for u\* does not include this shouldering reducing the variance of the u\* Gaussian PDF for event 11. This should be discussed. The numerical simulations are interesting and illustrate the sensitivity of the impact kinetic energy to different parameters on which the stability conditions could act. They clearly suggest that larger variances in u\* PDF, as generally observed in unstable conditions, generate stronger saltation and thus should be responsible for higher production of fine particles.*

In the very earlier phase of Event 11, there are some big $u_*$ values which produced the "shouldering". It is not that Event 11 is generally a case with bimodal $u_*$ values.

**Reply to Review 3:**

*I read the manuscript from the perspective of a specialist in wind erosion and dust emission but a generalist to this specific focus of the manuscript. The authors provide a clear, logical development of the focuses of their manuscript on the different bases for explaining particle size distributions of emitted dust and the dependency of airborne dust PSD on atmospheric boundary layer (ABL) stability. The topic is valuable for the community and the work is well presented. However, I am not convinced by the approach used in the manuscript. I think the work in the manuscript omits uncertainty. If that uncertainty were included, I think it may lead to different / alternative conclusions.*

*Therefore, to increase confidence in the results I think the omitted uncertainty must be tackled, in some form or other, before the work can be published. I provide below additional information on this point. I also think that some improvements in the structure of the manuscript will help the reader more easily follow the explanation of the work. In short, I am thoroughly supportive of the work. I think the manuscript needs to be revised to give confidence that the results are indeed detectable and therefore interpretations are robust to the uncertainty. The nature of the revisions I describe below I think, are consistent with a major revision, despite not being too difficult to achieve in a short period of time, if all other things were equal.*

Thanks for Review 3 for the support and the valuable comments. We fully agree, uncertainty analysis is important. In the revised paper, we have made the effort to increase the uncertainty analysis by (1) reworked on the full data set, this substantially increased the sample size; (2) added error bars in some graphs. We did not add error bars to all graphs, because the overlap of the error bars can make the graphs messy; (3) added a discussion section. The qualitative conclusion of the revised paper remains the same as in the earlier version.

*Main issues*

*Wind friction velocity uncertainty. In the abstract, it is stated that friction velocity u\* is a surrogate for surface shear stress and descriptor for saltation bombardment intensity. Line 32 states that "for a given soil, the particle size distribution of dust at emission (emission-dust PSD), ps(d), must depend on saltation bombardment or on friction velocity". The JADE field measured data are used to show that the (finely resolved) particle size distribution is dependent on measured wind friction velocity. In contrast to this approach, it is well known (cf. sediment transport models) that the wind energy available for saltation bombardment is not u\*, it is the energy us\*, which remains after wind momentum has been extracted by the roughness 'canopy'. In other words, us\*=u\*.R where R is the partition of drag. Under controlled conditions with homogeneous material, smooth surface (bed) without ripples and the bed reset after each experiment, it may be reasonable to assume that R=1. However, the authors use field data which, even under the smoothest field conditions are very likely to cause R not equal to 1. For example, soil has different sized aggregates at the surface, stones occur in the field, plant residue may be fixed to, or lying on, the soil and ripples may occur intermittently during sediment transport (and that is to say nothing of intermittent crusts which may change roughness). The magnitude of R<1 (which may change over time, between events due to change in the roughness 'canopy') is the omitted uncertainty in the authors' methodology.*

*For clarity, I think the authors should state clearly that they are assuming u\*=us\*. I think the authors must then ideally estimate, or at least approximate, the uncertainty of making that assumption (R=1) under field conditions when R<1. That 'model' uncertainty will then manifest as an error on u\*. When the PSDs are aggregated under this explicit approach, that 'model' uncertainty (not to be confused with the standard deviation of u\* already included by the authors) will demonstrate the extent to which there is a difference in the PSDs which is detectable. Any difference between the PSDs remaining after that 'model' uncertainty has been included will have accounted for the dependency of PSD on us\* and R.*

*I think the same issue of uncertainty occurs with the relation between the dependency of emission-dust PSD on u\* and the boundary-layer stability. Where u\* is based on field measurements, I think it requires the same (as above) expression of uncertainty. As above, this uncertainty is required due to the assumption that u\*=us\* when field conditions introduce uncertainty. Consequently, the results in the second half of the paper need to be similarly qualified with this uncertainty. The issue is brought to sharp focus by considering Eq. 8 of the manuscript. The sediment transport Q is related incorrectly to u\*^3 (Webb et al., 2020). As above, the available energy for transport is us\*^3. Whilst there are conditions when u\*=us\* and therefore R=1, in the field it is very unlikely that R=1. In this case, the uncertainty of the 'model' assumption u\*=us\*, needs to be considered (R<1). With this additional uncertainty the arising figures and interpretations may need to be re-evaluated.*

We are well aware of the excellent paper by Webb et al. (2020). However, to include the roughness related uncertainties would dramatically complicated the problem and will lead to more uncertainties in an already complex problem. Having said this, we believe the roughness issue is not so relevant to this study, because the JADE site was practically bare. What could influence roughness length are ripples. Using field measurements to detect changes in surface roughness (e.g. ripples) is difficult and in JADE we did not measure ripple changes. As saltation flux used in this study is measured, we do not see how the roughness correction comes into play. What can be done are probably some sensitivity tests on how erosion modified aerodynamic roughness length causes different saltation intensity, but this is already done in the recent paper by Webb et al. (2020). We have therefore not included the discussion on aerodynamic roughness in our paper.

*Manuscript structure*

*I think the manuscript mixes unnecessarily theory with results. I think the theory (Eqs. 4, 5, 6 etc. and related text) should be moved to the Methods section. In that Methods section I think it would be worthwhile explaining carefully how the parameter values of the modelling were chosen (rationale and assumptions) so that it is clear to the reader how the results have been produced. I find it strange not to have a Discussion section. I wonder if much of the detail in the Introduction would be better moved to the Discussion and then extended as necessary with additional context for the discussion. This would also help the Introduction quickly move the reader through the main issue.*

We have now substantially reworked on the manuscript. We hope that the views of Referee 3 are properly reflected in the revised paper.

**Reply to Dr. Sylvain Dupont**

We are most grateful to Dr. Sylvain Dupont for carefully reading our manuscript and providing insightful and detailed comments for discussion. These comments are very valuable for improving our paper.

Dr. Dupont first commented on the results of the Khalfallah et al. (2020) paper and pointed out possible discrepancies between his own analysis and the results presented in Khalfallah et al. Indeed, our paper is triggered by the study of Khafallah et al. which helped us decide to finally have a thorough look at the PSD issue. We thought this issue was resolved until the questions raised by the Kok paper which has caused a stir in the dust research community. Now, our results show the PSD at dust emission is $u_*$ dependent. This seems also to be the view of Dr. Dupont. With respect to PSD dependency on atmospheric boundary-layer stability, our results seem to support the finding of Khalfallah et al. qualitatively, but we have some considerations of their interpretation why this might be so. We are not convinced that "diffusion" caused this dependency.

As we do not know exactly, how colleagues Khalfallah et al. processed their data, we cannot judge the reliability of their conclusion. Dr. Dupont is in a much better position to make the judgement, as he works with the authors of the afore-mentioned paper. With the insight Dr. Dupont provided, we modified in the revised paper and to be more cautious with the interpretation of the results of Khalfallah et al., although we have certainly tried not to "rely" on their work.

The second point of Dr. Dupont is important, namely, that he is convinced of PSD dependency on $u_*$, not necessary on ABL stability. Our view is somewhat different. In our paper, we have tried to make it clear that there is a mean $u_*$ and a $u_*$ variance, the PSD of dust at emission is not only dependent on the $u_*$ mean but also on the $u_*$ variance. This dependency arises because the saltation bombardment is non-linearly dependent on $u^*$. In essence, this is the problem of saltation/dust emission intermittency. In a series of related studies (e.g. Klose et al. 2014), we have been considering how turbulence causes dust emission. Suppose the mean $u_*$ is $u_{*t}$, then there would be no saltation and no saltation bombardment, but if $u_*$ has a distribution, then intermittent saltation and saltation bombardment will occur, and the PSD of dust at emission will be dependent on the PDF of $u_*$. This really is the main point of this study, and the idea is already in several Klose et al. papers.

Now, does turbulence intensity (actually the PDF of $u_*$) depends on ABL stability? We think so, as the large-eddy simulation of Klose et al. (2014) shows and also the JADE observations. We have carried out recently a wind-tunnel experiment, again showing the dependency of $u_*$ PDF on ABL stability and the strong impact on dust emission. The results of the wind-tunnel experiments will be summarized and send for consideration of publication in our next paper on the subject.

Thanks for mentioning the Kaimal and Finnigan (1994) book (Yaping Shao and John Finnigan have worked in the same group for some years and is one of the first readers of the book). The $r_{uw}$ curve in Fig 1.9 of KJ (1994) book does not seem to apply here, because (1) there is nothing said about the variance of the shear stress only the mean; (2) it only states that shear stress normalized with the wind variances is fairly constant (not exactly constant, actually why not exactly?); (3) earlier measurement of shear stress was mainly down somewhere in the ABL at some level, not really close enough to the surface; and (4) their Fig 1.10 actually shows that the variance of wind is dependent on ABL stability (i.e. the variance normalized with $w_*$ is fairly universal).

We agree with Dr. Dupont, and we need to do more cases, as Ref. 2 also mentioned. In the revised paper, we used all JADE data and have done more case studies.

Dr. Dupont made several very helpful suggestions.

(1)     more figures for characterizing the events: to understand what happened during the erosion events 10 and 11, show time variation of dust and saltation PSD during the events.

This is a good idea. We have done more data analysis.

(2)     PSD of emitted dust flux.

We have added emission-flux PSD. In the revised text, we clarified the differences between the various definitions of PSDs.

(3)     Condensation

As far as we know, there was no condensation, but there were a few drops of rainfall accompanied with the cool change, although no rainfall was recorded. We have looked into this and have addressed this problem in the revised manuscript.

(4)     Enhanced cohesion in night.

This is an interesting point, but it is difficult to validate. However, we included Event-12 in the new study, which shows cohesion plays a big role in dust PSD.

In a separate study by the first author (unpublished), the modelling of soil moisture under extremely dry conditions is been worked out.

(5)     Surface modified by saltation.

Yes. This is likely, but we cannot validate this.

(6)     First justifications.

Dr. Dupont is right. We changed the manuscript.

(7)     $u_*$ variance.

This is an important point. As Dupont et al. (2018) shows that u* needs to be averaged over 15 to 30 min for the flux-gradient relationship. However, there is no doubt that shear stress fluctuates due to large eddies, and shear stress has a pdf. This pdf is important to dust emission which rapidly responses to surface shear stress. The selection of 1min for shear stress averaging seems to be reasonable, this is to assume that saltation can reasonably respond to shear stress variations on this scale. This is the whole point of this paper. We are willing to debate with Dr Dupont on this in greater detail. In the revised manuscript, we added a discussion section addressing some of the issues related.

(8)     Saltation bombardment intensity.

We have discussed this above. KF (1994) book, Fig. 1.9 states $r_{uw}$ is almost independent of z/L, but $r_{uw}$ is shear stress normalized with flow velocity variance which varies strongly with stability, as their Fig. 1.10 shows. But we agree with Dr. Dupont, this is an unsolved issue, because we do not fully understand how the laminar layer close to the surface behaves. There are theories about the possibility that the laminar layer breaks up. Our unpublished wind-tunnel experiment (measuring shear stress using Irwen sensors showing the fluctuations of shear stress related to large eddies). Again, the whole point is that we have to move away from the tradition "mean" flow approach and consider more the PDF of the turbulence quantities, which are important to understanding the PSD of dust at emission.

There seems to be a misunderstanding somewhere. What we try to say in justification 3 (now the second) is actually that the diffusion aspect due to enhanced or not enhanced turbulence with respect to instability does not affect the saltation trajectory too much. In this sense, Dr. Dupont is right. But the initial velocities of the saltation particles seem to be important.

It is great to discuss with colleague Dr. Dupont.

**Reply to Dr. Jasper Kok**

We greatly appreciate Colleague Dr. J. Kok for his comments. That Dr. Kok took time to provide such thoughtful comments shows the need to clarify the dust PSD issue.

First, "Airborne-dust PSD" as "emission-dust PSD": to our best knowledge, "emission-dust PSD" has never been directly observed. All emission-dust PSDs reported are airborne-dust PSDs or derived from airborne-dust PSDs. The various dust PSDs are sometimes confused in the literature. In the revised paper, we explicitly discussed this problem. The JADE airborne dust PSDs are of good quality and are probably close(r) to dust emission PSD.

The argument that dust advection depends on $u_*$ is interesting, but does not seem to apply here. Advection is $\sim u \, \partial C / \partial x \sim u_* \, \partial C / \partial x$. In case of weak dust concentration gradient, advection does not play a major role. The JADE site is fairly homogeneous and the dust PSDs are measured close to the surface. Therefore, we can safely exclude the influence of advection on dust PSD.

Second, "Consistency of Evidence". We have now checked this. The results presented in Shao et al. (2011) is based on 3.5m-OPC airborne-dust PSD for Event-10 and the results are correct. Colleague Dr. Kok and others have made excellent suggestions, so we have looked into the entire JADE dataset. Event-10 stands out as the only case, when dust PSD shows no clear u* dependency. This is interesting. We have now pointed this out in the paper and have provided some interpretations.

Statistical significance test is generally lacking in dust related studies and this is partially why we have so much confusion in aeolian research. We have now showed a lot of data and added a discussion section dedicated to the uncertainties of the analysis.

Third, earlier results: We like this suggestion of Dr. Kok very much. But, to be honest, this is difficult, as it is hard to get to the bottom of the various data sets. We believe Dr. Kok and colleagues have properly estimated the error margins of the previously published data. However, based on Kok, 2011b Table S1, the data used seem to be airborne-dust PSD. Also, the data shown in Figure 4 of Mahowald et al. 2013 (doi.org/10.1016/j.aeolia.2013.09.002) are mostly airborne-dust observations (mixed with emission-flux PSDs). We have now show that airborne-dust PSDs have height dependency. A proper scaling of dust concentration profile will be necessary, before all these profiles can be compared.

Fourth, statistics: This is a very good suggestion. We added error margins and sample sizes in the graphs.

5th Line 30-32: "Since inter-particle cohesion depends on particle size, d, the fraction of dust emitted must also depend on d. Thus, for a given soil, the particle size distribution of dust at emission (emission-dust PSD), ps(d), must depend on saltation bombardment or on friction velocity" and line 140-1 "u* is a descriptor of saltation bombardment intensity". This argument implicitly assumes that the impact speed of saltating particles increases with the friction velocity. It is highly intuitive that it would, but there is a very solid body of research that indicates that particle impact speed actually does not depend on friction velocity for transport-limited saltation. This lack of dependence of particle speed on wind speed was first proposed by Ungar and Haff (1987) because particle-wind feedbacks force an approximately constant saltator impact speed. It has since been confirmed by a large body of experimental (e.g., Namikas (2003), Rasmussen and Sorensen (2008), Creysells et al. (2009), Ho et al., (2011), Martin and Kok (2017)) and numerical (e.g., Duran et al. (2011), Kok et al. (2012)) work. The authors can of course present evidence to support their viewpoint counter to this literature, but I recommend acknowledging this extensive literature.

This is interesting. Let us make two thinking experiments. Exp 1: u* = u*t, particle creeps and has impact velocity 0. Exp 2: u* > u*t, particle saltates and has impact velocity larger than 0. This shows particle impact depends on u*. But thanks for pointing out the study which conclude differently. Nevertheless, in very strong saltation and very weak surface binding, it is possible that dust PSD dependency on u* is less obvious, as it seems to be case for Event-10.

6th Line 48-9: "Kok (2011a, 2011b) then proposed an emission-dust PSD and estimated its parameters from airborne-dust PSDs." That's actually not quite correct: Kok (2011a) only used emitted dust size distribution because airborne-dust PSDs are a convoluted sum of emission and advection (see comment above and by Sylvain Dupont). Also, the years on the references are incorrect (I corrected them in the quote above).

As far as we can see, Kok (2011a) used airborne-dust PSD.

7th I'm a bit confused how to interpret the 0-0.25 m/s u* category in the present paper's Figure 3, as this would include events without saltation where dust is not actively emitted but only advected. I suspect the authors are only using data for which saltation was occurring. If so, I recommend that the authors note that. And if not, I recommend the authors subset the data to only include active saltation data.

Saltation is intermittent and occurs below 0.25 m/s u*. This is a point we try to make, namely, turbulence (and saltation intermittency) plays an important role in dust PSD. It seems that this point did not come cross clearly, as this also appears to be the impression of colleague Dr. Dupont. However, we have followed the suggestion of Dr. Kok and have tested to average dust PSDs conditioned with $Q > 0.1$ gm$^{-1}$s$^{-1}$, but the results are almost the same.

Many thanks to Colleague Dr. J. Kok.

**List of Changes Made to the Manuscript:**

(1) Substantial rework on the JADE data. We now use the whole JADE dataset for the analysis (2) Added Event-10, Event-11 and Event-12 as case study (3) Redone most of the graphs (4) Added a section for discussion (5) Clarified a number of issues raised by the referees, Dr. Dupont and Dr. Kok (6) Clarified the definitions of emission-dust PSD, airborne-dust PSD and emission-flux PSD

(7) Pointed out the dependency of airborne-dust PSD on height (8) Pointed out that emission-flux PSD representing the dependency of dust concentration gradient on particle size (9) Added error-bars and sample size in graphs deemed to be useful (10) Numerous minor improvements

Please refer to the marked version for modification details.

[revised manuscript text omitted]